# ROBOT FLEET LEARNING VIA POLICY MERGING

**Lirui Wang[1], Kaiqing Zhang[2], Allan Zhou[3], Max Simchowitz[1], Russ Tedrake[1]**
[1]MIT CSAIL    [2]University of Maryland, College Park    [3]Stanford

## ABSTRACT

Fleets of robots ingest massive amounts of heterogeneous streaming data silos generated by interacting with their environments, far more than what can be stored or transmitted with ease. At the same time, teams of robots should co-acquire diverse skills through their heterogeneous experiences in varied settings. How can we enable such fleet-level learning without having to *transmit* or *centralize* fleet-scale data? In this paper, we investigate policy merging (PoMe) from such distributed heterogeneous datasets as a potential solution. To efficiently merge policies in the fleet setting, we propose FLEET-MERGE, an instantiation of distributed learning that accounts for the permutation invariance that arises when parameterizing the control policies with recurrent neural networks. We show that FLEET-MERGE consolidates the behavior of policies trained on 50 tasks in the Meta-World environment, with good performance on nearly all training tasks at test time. Moreover, we introduce a novel robotic *tool-use* benchmark, FLEET-TOOLS, for fleet policy learning in compositional and contact-rich robot manipulation tasks, to validate the efficacy of FLEET-MERGE on the benchmark.[1]

## 1 INTRODUCTION

With the fast-growing scale of robot fleets deployed in the real world, learning policies from the diverse datasets collected by the fleet (Osa et al., 2018; Bagnell, 2015; Kumar et al., 2020; 2022) becomes an increasingly promising approach to training sophisticated and generalizable robotic agents (Jang et al., 2022; Levine et al., 2020). We hope that both the magnitude of the data — streamed and actively generated by robots interacting with their surroundings — and the diversity of the environments and tasks around which the data are collected, will allow robot fleets to acquire rich and varied sets of skills. However, the data heterogeneity and total amount of the data are becoming as much of a challenge as a benefit. Real-world robot deployments often run on devices with real-time constraints and limited network bandwidth, while generating inordinate volumes of data such as video streams. Hence, a "top-down" scheme of *centralizing* these data (Grauman et al., 2022; Collaboration, 2023), and training a single policy to handle all the diverse tasks, can be computationally prohibitive and violate real-world communication constraints. At the same time, we wish to consolidate the skills each robot acquires after being trained on its local datasets via various off-the-shelf robot learning approaches. Thus, it is natural to ask: *How can the entire fleet efficiently acquire diverse skills, without having to* transmit *the massive amount of heterogeneous data that is generated constantly in silos, when each one of the robots has learned some skills from its own interactions?*

To answer this question, we propose *policy merging* (Figure 1), *PoMe*, a "bottom-up" approach for fleet policy learning from multiple datasets. Specifically, we consider neural-network-parameterized policies that are already *trained separately* on different datasets and tasks, and seek to merge their *weights* to form one single policy, while preserving the learned skills of the original policies. Policy merging acquires skills efficiently with drastically reduced communication costs, by transmitting only the *trained* weights of neural networks but not the training data. Such a bottom-up merging scheme is agnostic to and thus compatible with any local training approaches used in practice.

Merging the weights of neural networks has been studied in various contexts, including finetuning foundation models (Wortsman et al., 2022a;b), multi-task learning (He et al., 2018; Stoica et al., 2023), and the investigation of linear mode connectivity hypothesis for (feedforward) neural networks (Frankle et al., 2020; Entezari et al., 2021; Ainsworth et al., 2022). Distributed and federated learning

---

[1]Code is available at https://github.com/liruiw/Fleet-Tools.

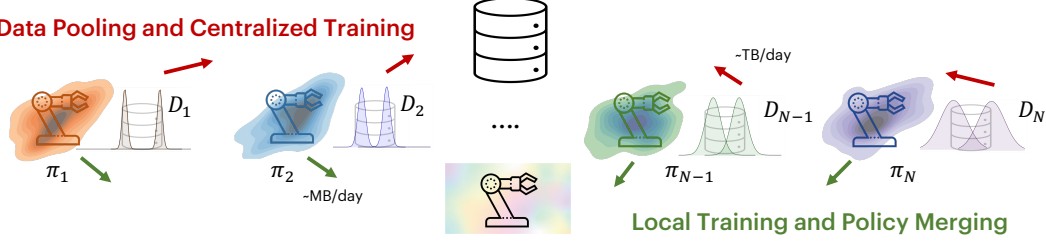

Figure 1: We consider the problem of merging multiple policies trained on potentially distinct and diverse tasks, which can be more computation and communication efficient than pooling all data together for joint training. Instead of acquiring astronomical size of data from the *top-down* (red arrow, requiring terabytes-per-day worth of data transfer), we demonstrate that the *bottom-up* approach (green arrow, megabytes-per-day): merging from locally trained policies, can also produce general policies that incorporate skills learned by the individual constituent policies. Moreover, local training and sharing weights are more suitable for agents that actively generate data, which is especially the case in robotic control, and are more efficient in communicating with the other agents. We aim to achieve the following objective in fleet learning: *One robot learns, the entire fleet learns.*

(McMahan et al., 2017; Wang et al., 2020a; Konečný et al., 2016), which iteratively update central model weights from decentralized updates, can be viewed as an *iterative* form of model merging; these approaches have achieved tremendous success in learning from diverse datasets, especially in solving *supervised learning* tasks (McMahan et al., 2017; Wang et al., 2020a; Mansour et al., 2020). However, such an approach has not yet demonstrated its power in solving *robot learning and control* tasks, which are generally more challenging due to their *dynamic* and *sequential* nature, and the richness of the tasks and environments. Notably, with the commonly used sensors in robot learning, e.g., cameras, one has to handle the partial observability in learning such *visuomotor policies* (Levine et al., 2016), which necessitates the use of *latent state* in parameterizing the policies, and is usually instantiated by recurrent neural networks (RNNs) (Elman, 1990). Indeed, policies with latent state dynamics are known to be theoretically necessary for partially-observed linear control problems (Bertsekas, 2012), and outperform policies parameterized by stateless policies in practice, i.e., those using feedforward neural networks, in perception-based robotic tasks (Andrychowicz et al., 2020).

In this work, we address policy merging in robotic fleet learning, with a focus on RNN-parameterized policies. A naive approach of averaging the weights would easily fail because multiple configurations of network weights parametrize the same function. One reason behind this is the known *permutation invariance* of neural networks, i.e., one can swap any two units of a hidden layer in a network, without changing its functionality (Hecht-Nielsen, 1990; Entezari et al., 2021). We have to account for such invariance in merging multiple policies. Indeed, such a fact has been accounted for recently in merging the weights of trained neural networks, with extensive focuses on aligning the weights of *feedforward* neural networks, and solving supervised learning tasks (Entezari et al., 2021; Ainsworth et al., 2022; Peña et al., 2022). We generalize these insights to the RNN setting, where permutation symmetries not only appear between *layers*, but also between *timesteps*, in solving robotic control tasks. Compared with one of the few federated learning methods that also explicitly account for permutation symmetry in merging RNNs (Wang et al., 2020a), we develop a new merging approach based on "soft" permutations (see Section 3 for a formal introduction) of the neurons, with more efficient implementation and an application focus on robotic control tasks. We detail our main contributions as follows, and defer a detailed related work overview in Appendix A.

- We design a new policy-merging approach, FLEET-MERGE, that outperforms baselines by over 50%, by accounting for the permutation symmetries in RNN-parameterized policies, and also extend the approach to the training stage, by allowing *multiple rounds* of merging between each training update, and also extend to merging *multiple (more than two)* models.
- We evaluate our proposed approach with different input modalities such as states, images, and pointclouds, in linear control and the Meta-World benchmark (Yu et al., 2020) settings.
- We develop and evaluated on a novel robotic tool-use benchmark, FLEET-TOOLS, for policy merging and fleet robot learning in compositional and contact-rich manipulation tasks.

## 2 PRELIMINARIES

**Model.** We consider the general setting of policy learning for controlling a dynamical system, when the system state may not be fully observed. Specifically, consider a sequential decision-making setting with time index $t \geq 1$, where at time $t$, a robot makes observations $o_t \in \mathcal{O}$ of a latent state $s_t \in \mathcal{S}$, and then selects a control action $a_t \in \mathcal{A}$. Dynamics and observations are characterized by probability distributions $\mathcal{P}_{\text{obs}}$ and $\mathcal{P}_{\text{dyn}}$, such that $o_t \sim \mathcal{P}_{\text{obs}}(\cdot \mid s_t)$ and $s_{t+1} \sim \mathcal{P}_{\text{dyn}}(\cdot \mid s_t, a_t)$. In the special case with full state observability, we have $\mathcal{O} = \mathcal{S}$ and $o_t = s_t$. In the special setting of *linear* control, both the dynamics and observations can be characterized by linear functions (with additional noises), see Appendix G for a formal and detailed introduction.

**Feedforward and recurrent policies.** For simplicity, we consider the case where the robot agent executes deterministic *recurrent* policies $\pi$. Specifically, we parameterize these policies by maintaining a policy *state* $h_t$, updated as $h_t = \pi(o_t, h_{t-1})$. As a special case, we consider *static feedback* policies $\pi : \mathcal{O} \to \mathcal{A}$ that select $a_t = \pi(o_t)$ as a function of only the instantaneous observation. Given a non-linear activation function $\sigma(\cdot)$, which is applied to vectors entry-wise, we can parameterize the static feedback policies by an $L$-layer *feedforward* neural networks with parameter $\theta = (\mathbf{W}_{\text{ff}}^\ell, \mathbf{b}^\ell)_{0 \leq \ell \leq L-1}$, where $\mathbf{W}_{\text{ff}}^\ell$ and $\mathbf{b}^\ell$ denote the weight and bias at layer $\ell$, respectively. The action $a_t = \pi_\theta(o_t)$ given observation $o_t$ is then given by

$$h^0 = o_t, \quad a_t = h^L, \quad h^{\ell+1} = \sigma\left(\mathbf{W}_{\text{ff}}^\ell h^\ell + \mathbf{b}^\ell\right), \quad 0 \leq \ell \leq L-1, \tag{2.1}$$

where above $h^\ell$ are the hidden layer activations. For the general case with recurrent policies, we parameterize them with Elman recurrent neural networks (Elman, 1990), with parameter $\theta = (\mathbf{W}_{\text{rec}}^{\ell+1}, \mathbf{W}_{\text{ff}}^\ell, \mathbf{b}^\ell)_{0 \leq \ell \leq L-1}$. Let $h_t = (h_t^\ell)_{1 \leq \ell \leq L}$ be a sequence of hidden states, such that $h_t = \pi_\theta(o_t, h_{t-1})$ is given by

$$h_t^0 = o_t, \quad a_t = h_t^L, \quad h_t^{\ell+1} = \sigma(\mathbf{W}_{\text{rec}}^{\ell+1} h_{t-1}^{\ell+1} + \mathbf{W}_{\text{ff}}^\ell h_t^\ell + \mathbf{b}^\ell), \quad 0 \leq \ell \leq L-1. \tag{2.2}$$

The presence of the recursive term $\mathbf{W}_{\text{rec}}^{\ell+1} h_{t-1}^{\ell+1}$ that incorporates the hidden state at the previous time $t-1$ distinguishes the RNN architecture in Eq. (2.2) from the feedforward architecture in Eq. (2.1). Notice here that at time $t$, the action $a_t$ is just the last layer of the hidden state $h_t^L$.

**Permutation invariance.** As well-understood for supervised learning (Frankle et al., 2020; Wortsman et al., 2022b; Ainsworth et al., 2022), policies trained separately – even on the same dataset – can exhibit similar behavior whilst having very different weights. This is in large part due to the *invariances* of neural network architectures to symmetry transformations. For an $L$-layer neural network with layer-dimensions $d_0, d_1, \ldots, d_L$, let $\mathcal{G}_{\text{perm}}$ denote the set of *hard permutation operators*. These are sequences of matrices $\mathcal{P} = (\mathbf{P}^0, \mathbf{P}^1, \ldots, \mathbf{P}^L)$, where we always take $\mathbf{P}^0 = \mathbf{I}_{d_0}, \mathbf{P}^L = \mathbf{I}_{d_L}$, and take $\mathbf{P}^\ell$ as a $d_\ell \times d_\ell$ *permutation* matrix for $1 \leq \ell \leq L-1$. We let $\mathcal{G}_{\text{lin}} \supset \mathcal{G}_{\text{perm}}$ denote the set of *linear transformation operators* that are sequences of matrices $\mathcal{P} = (\mathbf{P}^0, \mathbf{P}^1, \ldots, \mathbf{P}^L)$, where we still have $\mathbf{P}^0 = \mathbf{I}_{d_0}, \mathbf{P}^L = \mathbf{I}_{d_L}$, but now we allow $(\mathbf{P}^1, \ldots, \mathbf{P}^{L-1})$ to be general *invertible* matrices. Elements of $\mathcal{G}_{\text{lin}}$ (and thus $\mathcal{G}_{\text{perm}}$) act on feedforward models $\theta$ via

$$(\mathbf{W}_{\text{ff}}^\ell, \mathbf{b}^\ell) \mapsto (\mathbf{P}^{\ell+1} \mathbf{W}_{\text{ff}}^\ell (\mathbf{P}^\ell)^{-1}, \mathbf{P}^{\ell+1} \mathbf{b}^\ell). \tag{2.3}$$

It is known that the feedforward architecture Eq. (2.1) is invariant, in terms of input-output behavior, to all hard permutation transformations $\mathcal{P} \in \mathcal{G}_{\text{perm}}$, but not to general $\mathcal{P} \in \mathcal{G}_{\text{lin}}$. When the activation function is an identity mapping, i.e., the neural networks are linear, it becomes invariant to $\mathcal{G}_{\text{lin}}$.

**Imitation learning.** As a basic while effective imitation learning method, we here focus on behavior cloning (Osa et al., 2018; Bagnell, 2015) for the purpose of introducing the policy-merging framework next. Note that our merging framework and algorithms will be agnostic to, and can be readily applied to other imitation learning algorithms. In behavior cloning, one learns a policy $\pi_\theta$, parameterized by some $\theta \in \mathbb{R}^d$, that in general maps the observation-action trajectories to actions, by imitating trajectories generated by expert policies. Let $\mathcal{D} = (\boldsymbol{\tau}^{(i)})_{1 \leq i \leq M}$ denote a set of $M$ trajectories, with $\boldsymbol{\tau}^{(i)} = (o_t^{(i)}, a_t^{(i)})_{1 \leq t \leq T}$ denoting the $i$-th trajectory of length $T$. As an example, we study behavior cloning with the $\ell_2$-imitation loss, instantiated by $\bar{\mathcal{L}}_{\text{bc}}(\theta; \mathcal{D}) := \sum_{i=1}^M \mathcal{L}_{\text{bc}}(\theta; \boldsymbol{\tau}^{(i)})$, where for a given $\boldsymbol{\tau} = (o_t, a_t)_{1 \leq t \leq T}$,

$$\mathcal{L}_{\text{bc}}(\theta; \boldsymbol{\tau}) := \sum_{t=1}^T \|\hat{a}_{\theta,t} - a_t\|^2, \quad \text{where } \hat{h}_{\theta,t} := \pi_\theta(o_t, \hat{h}_{\theta,t-1}), \quad \hat{a}_{\theta,t} = \hat{h}_{\theta,t}^L, \tag{2.4}$$

where $\hat{h}_{\theta,t}$ denotes the hidden state that arises from executing the recurrent policy $\pi_\theta$ on the observation sequence $o_1, o_2, \ldots, o_t$, via the Elman recurrent updates in Eq. (2.2), and the action is part of the hidden state corresponding to the last layer. Note that in the special case where $\pi_\theta$ is a static feedback policy, we can drop $\hat{h}_{\theta,t}$ from the above display, and generate each $\hat{a}_{\theta,t}$ using $o_t$ based on Eq. (2.1).

**Policy merging framework.** We now introduce *policy merging*, our framework for fleet policy learning from diverse datasets. Consider $N$ datasets collected by a fleet of $N$ robots from possibly different tasks and environments, and can potentially be highly heterogeneous and non-i.i.d. Each robot agent $i = 1, 2, \cdots, N$ only has access to the dataset $\mathcal{D}_i$ of itself, in the form given in Section 2. Ideally, if the robot designer can pool all the data together, then the objective is to minimize the following imitation loss across datasets

$$\min_\theta \quad \sum_{i=1}^N \bar{\mathcal{L}}_{\mathrm{bc}}(\theta; \mathcal{D}_i). \tag{2.5}$$

Let $\theta_{\mathrm{pool}}$ denote the solution[2] to Eq. (2.5), i.e., the best policy parameter one can hope for when seeing all the data, and will provide an upper bound for the performance we will compare with later. Let $\theta_i$ denote the policy parameter of robot $i$ by minimizing the loss associated with $\mathcal{D}_i$, i.e., $\theta_i \in \mathrm{argmin}_\theta \bar{\mathcal{L}}_{\mathrm{bc}}(\theta; \mathcal{D}_i)$. The goal of policy merging is to find a single policy parameter $\theta_{\mathrm{mrg}}$, as some aggregation of the local policy parameters $(\theta_1, \cdots, \theta_N)$, *without sharing the datasets*.

An example of this aggregation is *direct averaging*, i.e., $\theta_{\mathrm{mrg}} = \bar{\theta} := \frac{1}{N} \sum_{i=1}^N \theta_i$. We propose more advanced policy merging methods in Section 3 by accounting for the symmetries of RNN weights. Note that the above merging process can also occur *multiple* times during training: at round 1, we first merge the trained policies $(\theta_1^{(1)}, \cdots, \theta_N^{(1)})$ to obtain $\theta_{\mathrm{mrg}}^{(1)}$, and send it back to the robots to either conduct more training and/or collect more data. At round 2, the newly trained local policy parameters $(\theta_1^{(2)}, \cdots, \theta_N^{(2)})$ are then merged to obtain $\theta_{\mathrm{mrg}}^{(2)}$. This iteration can proceed multiple times, and we will refer to it as the *iterative merging* setting. When the merging is instantiated by direct averaging, this iterative setting exactly corresponds to the renowned *FedAvg* algorithm in federated learning (McMahan et al., 2017). When there is only one such iteration, we refer to it as the *one-shot merging* setting. Fewer iterations lead to fewer communication rounds between the robots and the designer, and note that no data is transmitted between them.

## 3 METHODOLOGY

Given the invariance properties introduced in Section 2, merging by naive averaging the parameters $\theta_{\mathrm{mrg}} \leftarrow \frac{1}{N} \sum_{i=1}^N \theta_i$ may not perform well. Prior work has instead proposed merging *aligned* feedforward neural network models $\theta_{\mathrm{mrg}} \leftarrow \frac{1}{N} \sum_{i=1}^N \mathcal{P}_i(\theta_i)$, where the elements $\mathcal{P}_1, \ldots, \mathcal{P}_N$ are weight transformations which are often, but not necessarily, hard permutation operators (i.e., elements of $\mathcal{G}_{\mathrm{perm}}$). The GITREBASIN algorithm (Ainsworth et al., 2022) iteratively computes the weight permutations $(\mathcal{P}_i)$. At each step, agent index $i$ is drawn uniformly from $[N]$, and one constructs $\theta_i' = \frac{1}{N-1} \sum_{j \neq i} \theta_j$ by averaging the parameters of indices $j \neq i$. It then solves a series of linear assignment problems (LAPs) (Kuhn, 1955; Jonker & Volgenant, 1988; Bertsekas, 1998) for each layer $\ell$ to find some $\mathbf{P}^\ell$, which is derived by matching the activations between two models via ordinary least squares regression. The algorithm then repeats the sampling from $(\theta_1, \ldots, \theta_N)$ and the computation of $\theta_i'$, until convergence.

Peña et al. (2022) instead propose a gradient-based variant to merge *two* models by relaxing the rigid constraint of using a *hard* permutation matrix. The direct extension of their algorithm to our setting is as follows: given two models $(\theta, \theta')$, iteratively trajectories $\tau$ from a common dataset $\mathcal{D}$, and update the aligning parameters $\mathcal{P}$ by following the gradient of $\mathcal{L}_{\mathrm{bc}}(\alpha\mathcal{P}(\theta) + (1-\alpha)\theta'; \tau)$, where $\mathcal{L}_{\mathrm{bc}}$ is as given in Eq. (2.4). Thus, for each iteration $s \geq 1$

$$\tilde{\mathcal{P}}_s \leftarrow \mathcal{P}_s - \eta \nabla_{\mathcal{P}} \mathcal{L}_{\mathrm{bc}}(\alpha\mathcal{P}(\theta) + (1-\alpha)\theta'; \tau)\big|_{\mathcal{P}=\mathcal{P}_s}, \quad \alpha \sim \mathrm{Unif}[0,1], \quad \tau \sim \mathcal{D} \tag{3.1}$$

with some stepsize $\eta > 0$. Note that the updated matrices in $\tilde{\mathcal{P}}_s = (\tilde{\mathbf{P}}_s^0, \ldots, \tilde{\mathbf{P}}_s^L)$ are not necessarily (even close to) permutation matrices. We define a *soft permutation projection* with regularization

---

[2]For convenience, we speak heuristically of exact minimizers in this section. In practice, we understand "minimizer" as "model trained to minimize the given loss".

---

**Algorithm 1** FLEET-MERGE: Fleet Learning of Policies via Weight Merging

---

1: **Input**: Models $\theta_1, \ldots, \theta_N$, datasets $\mathcal{D}_{\text{local},i}$ for each $i \in [N]$
2: **Parameters:** Epoch length $E$, iteration number $S$, soft-projection parameter $\tau > 0$, stepsize $\eta > 0$
3: **Initialize:** Permutations $\mathcal{P}_{\text{hard},1}, \ldots, \mathcal{P}_{\text{hard},N}, \mathcal{P}_{\text{soft},1}, \ldots, \mathcal{P}_{\text{soft},N} \leftarrow$ Identity
4: **for** Epoch $s = 1, \ldots, E$ **do**
5:   **Average** models: $\bar{\theta} \leftarrow \frac{1}{N} \sum_{i=1}^{N} \mathcal{P}_{\text{hard},i}(\theta_i)$
6:   **Sample** indices $\mathcal{I} \subset [N]$
7:   **for** $i \in \mathcal{I}$ **do**
8:     **for** Iteration $t = 1 \ldots, T$ **do**
9:       **Sample** data pair $(o, a) \sim \mathcal{D}_{\text{local},i}$ to form a trajectory $\boldsymbol{\tau}$, and sample interpolation parameter $\alpha \sim \text{Unif}[0,1]$
10:      **Update** with gradient: $\tilde{\mathcal{P}}_{\text{soft},i} \leftarrow \mathcal{P}_{\text{soft},i} - \eta \nabla_{\mathcal{P}} \mathcal{L}_{\text{bc}}(\alpha \mathcal{P}(\theta) + (1-\alpha)\bar{\theta}; \boldsymbol{\tau})\big|_{\mathcal{P} = \mathcal{P}_{\text{soft},i}}$
11:      **Update** $\mathcal{P}_{\text{soft},i} \leftarrow \text{Proj}_\tau(\tilde{\mathcal{P}}_{\text{soft},i})$ by applying the soft projection step Eq. (3.2)
12:      **Update** $\mathcal{P}_{\text{hard},i} \leftarrow \text{Proj}_{\text{hard}}(\mathcal{P}_{\text{soft},i})$
13: **Return**: $\bar{\theta} = \frac{1}{N} \sum_{i=1}^{N} \mathcal{P}_{\text{hard},i}(\theta_i)$

---

$\tau > 0$ as:

$$\text{Proj}_\tau(\tilde{\mathcal{P}}) = \mathbf{P}^{1:L-1}, \quad \text{where } \mathbf{P}^\ell \in \underset{\mathbf{P} \in \mathscr{B}_{d_\ell}}{\arg\max} \ \langle \tilde{\mathbf{P}}^\ell, \mathbf{P} \rangle_{\text{F}} + \tau \mathscr{H}(\mathbf{P}), \tag{3.2}$$

and the associated *hard permutation projection* $\text{Proj}_{\text{hard}} := \text{Proj}_\tau\big|_{\tau=0}$, where $\mathscr{B}_d$ is the Birkhoff polytope of doubly-stochastic matrices, $\mathscr{H}(\mathbf{P}) = -\sum_{i,j} \mathbf{P}_{ij} \log(\mathbf{P}_{ij})$ is the matrix entropy (Cuturi, 2013; Mena et al., 2018), and $\tau > 0$ is some hyperparameter that weights the strength of the entropy regularization. Computation of $\text{Proj}_{\text{hard}}$ can be implemented efficiently via solving a linear assignment problem, and the solution with $\tau > 0$ can be solved approximately via a Sinkhorn iteration (Eisenberger et al., 2022; Peña et al., 2022), which also allows gradient computation on any differentiable objective. The operators are then updated as $\mathcal{P}_{s+1} \leftarrow \text{Proj}_\tau(\tilde{\mathcal{P}}_s)$. Note that for $\ell = 0$ and $L$, we directly set $\mathbf{P}^\ell_{s+1}$ to be identity matrices. At the final step, Peña et al. (2022) conducts a *hard* projection onto the space of permutation matrices.

To measure the performance of model alignment, we study the *(imitation) loss barrier*, as defined previously in the supervised learning setting (Frankle et al., 2020; Ainsworth et al., 2022). Specifically, given two policy parameters $\theta, \theta'$ such that $\bar{\mathcal{L}}_{\text{bc}}(\theta; \mathcal{D}) \approx \bar{\mathcal{L}}_{\text{bc}}(\theta'; \mathcal{D})$, the loss barrier of the policies, defined as $\max_{\lambda \in [0,1]} \bar{\mathcal{L}}_{\text{bc}}((1-\lambda)\theta + \lambda\theta'; \mathcal{D}) - \frac{1}{2}(\bar{\mathcal{L}}_{\text{bc}}((\theta; \mathcal{D}) + \bar{\mathcal{L}}_{\text{bc}}(\theta'; \mathcal{D}))$, evaluates the worst performing policy linearly interpolating between $\theta$ and $\theta'$, where we recall the definition of $\bar{\mathcal{L}}_{\text{bc}}$ above Eq. (2.4). More amenable to the control and policy learning setting, we can also define the task *performance barrier*, which replaces the behavior cloning loss $\bar{\mathcal{L}}_{\text{bc}}$ with any suitable measure $\mathcal{T}$ of task performance (e.g., the accumulated rewards or the success rates of task completion): $\max_{\lambda \in [0,1]} \frac{1}{2}(\mathcal{T}(\theta) + \mathcal{T}(\theta')) - \mathcal{T}((1-\lambda)\theta + \lambda\theta')$; the sign is flipped to model the rewards achieved in accomplishing the tasks. These metrics will be used in our experiments in Section 5.

### 3.1 MERGING MANY RECURRENT POLICIES

In this section, we describe our new algorithm for merging many RNN-parameterized policies.

**Permutation invariance of RNNs.** Given a recurrent neural network with parameter $\theta = (\mathbf{W}_{\text{rec}}^{\ell+1}, \mathbf{W}_{\text{ff}}^\ell, \mathbf{b}^\ell)_{0 \leq \ell \leq L-1}$ as parameterized in Eq. (2.2), we let weight transformations $\mathcal{P} = (\mathbf{P}^0, \mathbf{P}^1, \ldots, \mathbf{P}^L) \in \mathcal{G}_{\text{lin}}$ act on it through

$$\left(\mathbf{W}_{\text{rec}}^\ell, \mathbf{W}_{\text{ff}}^{\ell-1}, \mathbf{b}^{\ell-1}\right) \mapsto \left(\mathbf{P}^\ell \mathbf{W}_{\text{rec}}^\ell (\mathbf{P}^\ell)^{-1}, \ \mathbf{P}^\ell \mathbf{W}_{\text{ff}}^{\ell-1} (\mathbf{P}^{\ell-1})^{-1}, \ \mathbf{P}^\ell \mathbf{b}^{\ell-1}\right) \tag{3.3}$$

for each layer $\ell$ with $1 \leq \ell \leq L-1$. In Appendix B, we verify that RNNs are invariant to the above operation when $\mathcal{P} \in \mathcal{G}_{\text{perm}} \subset \mathcal{G}_{\text{lin}}$ are hard permutation operators.

**Proposition 3.1.** Any recurrent neural network given by Eq. (2.2) is invariant to any transformation of $\mathcal{P} \in \mathcal{G}_{\text{perm}}$.

In Appendix B, we also expand upon the group structure of $\mathcal{G}_{\mathrm{perm}}$, to the larger sets of invariance groups, for the special architecture with ReLU activations. Moreover, we also argue that permutations are in essence the *only* generic invariances of ReLU and polynomial networks whose weights minimize the $\ell_2$ norm, which might be of independent interest.

**Merging many models with a single reference.** Rather than *sequentially* merging $N = 2$ models, we merge all models to a common reference model $\bar{\theta}$. Inspired by the update rules in federated learning, this approach has the following advantages: (a) it removes the dependence of sequential merging on the *order* of the merging sequence; (b) it allows better pooling of weights from *many* models to guide each merging step; (c) we align only a *subset* of models per iteration, which becomes more efficient when $N$ is very large. We show the benefits of our approach over sequential merging in ablation studies (Appendix F). In addition, unlike the *two-model-merging* setting of Peña et al. (2022), we do not have access to a common dataset $\mathcal{D}$ for the aligning updates. Instead, each model is updated by sampling trajectories from its local dataset $\mathcal{D}_{\mathrm{local},i}$, obviating the need for dataset sharing.

**Algorithm description.** Our algorithm, FLEET-MERGE, is depicted in Algorithm 1. We maintain hard transformation operators $\mathcal{P}_{\mathrm{hard},1}, \ldots, \mathcal{P}_{\mathrm{hard},N} \in \mathcal{G}_{\mathrm{perm}}$, initialized by identity matrices. At each epoch, we compute the reference model $\bar{\theta}$ by averaging each model under the associated transformation (Line 5). We then select a subset of models $\mathcal{I}$, and initialize the "soft" permutation $\mathcal{P}_{\mathrm{soft},i} \leftarrow \mathcal{P}_{\mathrm{hard},i}$ as the hard permutation operator. For each $i \in \mathcal{I}$, we update the "soft" permutation $\mathcal{P}_{\mathrm{soft},i}$ for $T$ steps. Importantly, because each $\theta_i$ corresponds to a recurrent neural network model, the action of $\mathcal{P}_{\mathrm{soft},i}(\theta_i)$ in the gradient step in Line 10 is given by Eq. (3.3). Of equal significance (and as noted above), the trajectories $\boldsymbol{\tau}$ in Line 9 are sampled not from a common dataset, but rather from a local dataset $\mathcal{D}_{\mathrm{local},i}$ associated with the $i$-th agent. We conclude by re-projecting each $\mathcal{P}_{\mathrm{soft},i}$ onto $\mathcal{G}_{\mathrm{perm}}$ to obtain a new $\mathcal{P}_{\mathrm{hard},i}$ (Line 12), which are used to update $\bar{\theta}$ accordingly in the next epoch. In particular, our algorithm allows *iterative merging*, i.e., merging multiple times during the course of training, compared to other (feedforward) neural network merging approaches (Ainsworth et al., 2022; Peña et al., 2022; Stoica et al., 2023), which allows tradeoffs between communication cost and merged-model performance. At test time, the merged policy $\pi_{\bar{\theta}}$ is deployed for the test task. In the multi-task setting, we consider the cases where the task identity can be inferred from observations.

## 4 FLEET-TOOLS: A ROBOTIC TOOL-USE SIMULATION BENCHMARK

To validate the performance of our policy-merging framework, we develop a new robotic tool-use benchmark, FLEET-TOOLS, in the robotic simulator Drake (Tedrake & the Drake Development Team, 2019). The benchmark FLEET-TOOLS focuses on robotic manipulation tasks with rich contact-dynamics, and category-level composition, as in several existing literature in robotic manipulation for tool-use (Toussaint et al., 2018; Holladay et al., 2019).

**Expert tasks.** To scale the generation of expert demonstrations for the tool-use tasks, we specify the tasks via keypoints (Manuelli et al., 2022) when generating the expert trajectories. We mainly consider four skills as examples (or task families): {`wrench`, `hammer`, `spatula`, and `knife`} – consisting of tasks involving the eponymous tool-type. Specifically, we consider `use a spanner to apply wrenches`, `use a hammer to hit`, `use a spatula to scoop`, and `use a knife to split`. All of these tasks can be solved to reasonable performance by specifying a few keypoints. For instance, in the `wrench` tasks as shown in Figure 5, the wrench needs to reach the nut first and then rotate by 45 degrees while maintaining contact. The same strategy can be applied to a set of nuts and wrenches. We choose these tasks because they provide a balance of common robotic tasks that require precision, dexterity, and generality in object-object affordance. We use the segmented tool and object point-cloud as the observation $o_t$, and the translation and rotation of the robot end effector (6-dof) as the action $a_t$. At test time of a multi-task setting, the task would be identified by the observations, i.e., the (combination of) a tool and an object, which can be inferred from the point could as the input of the test policy, identifies the task being tested on.

**Keypoint transformation & trajectory optimization.** We denote the end effector frame and world frame as $E$ and $W$, respectively. For each tool, we specify $n$ keypoints, for example, when $n = 3$, we can specify one point on the tool head, one point on the tool tail, and one point on the side. We define a rigid transformation $\mathbf{X} \in \mathrm{SE}(3) \subset \mathbb{R}^{4 \times 4}$, the current locations of the $n$ keypoints on the tool in the world frame $W$ as ${}^W\mathbf{p}^{\mathtt{tool}} = \left[ {}^W\mathbf{p}^{\mathtt{tool},1}, \cdots, {}^W\mathbf{p}^{\mathtt{tool},n} \right]$, and the poses in the

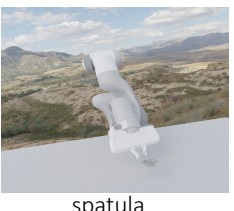 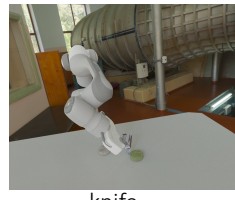 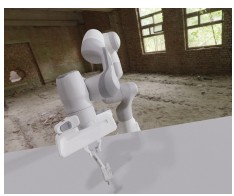 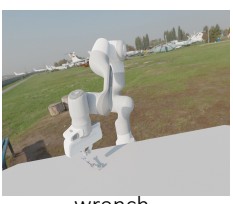

spatula              knife              hammer              wrench

Figure 2: FLEET-TOOLS Benchmark. We develop several tool-use tasks that focus on contact-rich motions and category-level compositions in the Drake simulator.

end-effector frame as $^E\mathbf{p}^{\texttt{tool}} = \left[^E\mathbf{p}^{\texttt{tool},1}, \cdots, ^E\mathbf{p}^{\texttt{tool},n}\right]$, respectively. For the object, we specify $m$ keypoints in the world frame as $^W\mathbf{p}^{\texttt{obj}} = \left[^W\mathbf{p}^{\texttt{obj},1}, \cdots, ^W\mathbf{p}^{\texttt{obj},m}\right]$. Then, one can solve an optimization problem, referred to as *kPAM-Opt* (where kPAM stands for *KeyPoint Affordance-based Manipulation*) (Manuelli et al., 2022), to find the transformation $\mathbf{X}$ (see Eq. (C.1)-Eq. (C.3) for the detailed formulation). However, recovering the joint angle from the transformation $\mathbf{X}$ via inverse kinematics can be computationally challenging. Hence, we propose to solve the kPAM optimization in the *joint* space, i.e., optimize over the joint angles $\mathbf{q}$ that parameterize the transformation through forward kinematics (i.e., `forward-kinematics(q)`), and only add constraints to the keypoints in the transformed frame. Specifically, we solve the following joint-space optimization problem:

$$\boxed{\text{Joint-Space-kPAM-Opt}} \qquad \min_{\mathbf{q}, \mathbf{X}} \qquad \left\|\mathbf{X}^E\mathbf{p}^{\texttt{tool}} - {}^W\mathbf{p}^{\texttt{tool}}\right\|_{\text{F}}^2 \tag{4.1}$$

$$\text{s.t.} \qquad \left\|\mathbf{X}^E\mathbf{p}^{\texttt{tool},i} - {}^W\mathbf{p}^{\texttt{obj},j}\right\|_2 \le \epsilon, \ \ \forall i, j \tag{4.2}$$

$$\alpha_{i,j} - \epsilon \le \beta_{i,j}^\top \mathbf{X}(^E\mathbf{p}^{\texttt{tool},i} - {}^E\mathbf{p}^{\texttt{tool},j}) \le \alpha_{i,j} + \epsilon, \ \ \forall i, j \tag{4.3}$$

$$\mathbf{X} = \texttt{forward-kinematics}(q) \tag{4.4}$$

where we choose $i \in \{1, \cdots, n\}$ and $j \in \{1, \cdots, m\}$, and $\beta_{i,j} \in \mathbb{R}^4$ is a vector on the unit sphere with the fourth dimension $\beta_{i,j}(4) = 0$, $\alpha_{i,j}$ is some constant that represents the target angle alignment, and $\epsilon > 0$ is some relaxation level. We provide a detailed explanation of the constraints in §C.1. After finding the optimal joint angle, one can find pre-actuation and the post-actuation trajectories by solving a trajectory planning problem (see Eq. (C.8)-Eq. (C.10) for more details).

The benchmark can be easily extended to new tool-use tasks by labeling a few keypoints and providing a configuration for constraints and costs for the desired task. The overall pipeline can also easily represent "category-level" composition and generalization, across various (combinations of) tools and objects, providing a systematic and scalable way to create "local and distributed demonstration datasets". This makes FLEET-TOOLS a great fit for evaluating large-scale fleet policy learning. More details of the benchmark can be found in Appendix C.

## 5 EXPERIMENTS: POLICY MERGING

We evaluate the performance of various algorithms in several benchmark environments: FLEET-TOOLS, as described in Section 4, Meta-World (Yu et al., 2020), which has 50 distinct robotic manipulation tasks, and linear control. This section summarizes results on FLEET-TOOLS and Meta-World, deferring additional experimental details with different network architectures and results to Appendix C.2 and Appendix D. Details for our results on linear control problems can be found in Appendix G. Ablation studies have also been conducted for supervised learning settings including classification (Deng, 2012) and language generation (Caldas et al., 2018); these are deferred to Appendix F. To summarize, the ablations validate the benefits of: (a) merging to a common reference model (Line 5) and (b) performing hard projections (Line 12) in Algorithm 1, as well as other algorithm-design choices.

**Baselines.** We compare our FLEET-MERGE against the following baselines {NAIVEAVERAGE, GITREBASIN, SINGLEDATASET}: NAIVEAVERAGE and GITREBASIN denote the method of naive averaging and that in Ainsworth et al. (2022), respectively, i.e., for NAIVEAVERAGE, it directly averages $\bar{\theta} = \lambda\theta + (1 - \lambda)\theta'$, and for GITREBASIN, it computes an aligning permutation $\mathcal{P}$ from the ReBasin algorithm in Ainsworth et al. (2022) and merges $\bar{\theta} = \lambda\theta + (1 - \lambda)\mathcal{P}(\theta')$. SINGLEDATASET trains on a single dataset which differs from the test dataset, and thus measures the performance of "zero shot adaptation" in contrast to model merging. Each merging method is applied *only once* after training is complete. We study how the performance of these methods varies as the *source distributions* of datasets on which the different models are trained become more diverse.

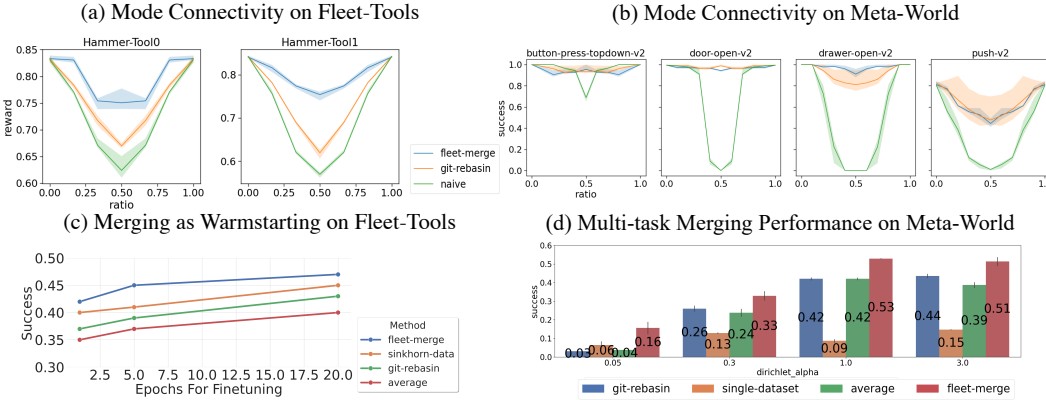

Figure 3: Results on One-Shot Merging. **(a & b)** FLEET-MERGE attains better mode connectivity for policies in both FLEET-TOOLS and Meta-World. **(c)** One-shot merging performance is predictive of relative performance after finetuning. **(d)** FLEET-MERGE succeeds in multitask settings with varying data heterogeneity for 5 data sources, as measured by the Dirichlet parameter $\alpha$.

**Data heterogeneity.** We also validate the performance of policy merging by varying the levels of *data heterogeneity*, an important metric in distributed and federated learning (McMahan et al., 2017; Hsieh et al., 2020). We adopt a popular approach in federated learning (Hsieh et al., 2020) to create data heterogeneity: for each experiment, we begin with $K$ component distributions $\mathcal{D}_1^\star, \ldots, \mathcal{D}_K^\star$ (e.g., imitation learning from a `hammer` task v.s. a `spatula` task). We then sample $N$ source distributions by sampling a mixture weights $p_1, \ldots, p_N \in \triangle(K) \sim \text{Dirichlet}(\alpha \mathbf{1}_K)$ from an evenly-weighted Dirichlet distribution with parameter $\alpha$ in dimension $K$. Smaller $\alpha$ biases toward "peakier" probability distributions, encouraging greater dataset heterogeneity. By default, we use $N = 5$ and each data source has the same number of data points.

## 5.1 ONE-SHOT MERGING

We first study one-shot model merging, and validate the presence of *mode connectivity* between the policy models, as in the weight-merging literature for feedforward NNs (Frankle et al., 2020; Entezari et al., 2021). We consider two policy models $\theta, \theta'$ trained on the *same* dataset, and measure the performance of NAIVEAVERAGE and GITREBASIN (as described above), and SOFTUPDATE, which computes the aligning perturbation using the soft-projection updates as described in Section 3.

**FLEET-TOOLS.** As a preliminary test, we first train multiple feedforward policies on a single dataset with distinct tasks with behavior cloning. In Figure 3 (a), we evaluate the performance barrier for every pair of the trained policies. We observe that the algorithm that accounts for permutation alignment has a significantly smaller performance barrier when interpolating along the segment in parameter space between two aligned weights. This implies that it is possible to acquire a policy by merging the policies trained on separate datasets, without sharing data or changing the input-output behavior of each individual model. We extend this merging setting to multiple datasets and multiple tasks. In Figure 3 (c), we show that the merged model can also serve as a warm start for policy finetuning, which can be further finetuned on downstream tasks with a held-out small dataset. In this case, FLEET-MERGE enjoys superior performance compared to naive averaging, git-rebasin Ainsworth et al. (2022)'s extension to multiple models, and Sinkhorn rebasin Peña et al. (2022)'s.

**Meta-World.** We use the frozen ResNet features on the images as policy inputs. In Figure 3 (b), we compare different merging algorithms by measuring the mode connectivity in the one-shot merging regime. We observe that there are almost *no barriers* for certain tasks. See Appendix D for more experiments with different network architectures. In Figure 3 (d), we show FLEET-MERGE outperforms baseline methods in the challenging multitask setting, across all settings of the Dirichlet parameter $\alpha$ (which, we recall, controls the local dataset heterogeneity).

**Linear control.** Lastly, we evaluate on Linear Quadratic Gaussian (LQG) control tasks with high-dimensional observations. We briefly summarize our findings here and defer a detailed description of the setting, algorithms, and experiments to Appendix G, together with the implications of policy merging on the topology of the LQG landscape. In short, we compare three baselines: direct averaging of policy parameters, permutation-based averaging computed via linear sum assignment, and a gradient-based alignment approach that aligns over all invertible matrices. First, we discover that our framework can still be effective for merging linear control policies, in both single-task and

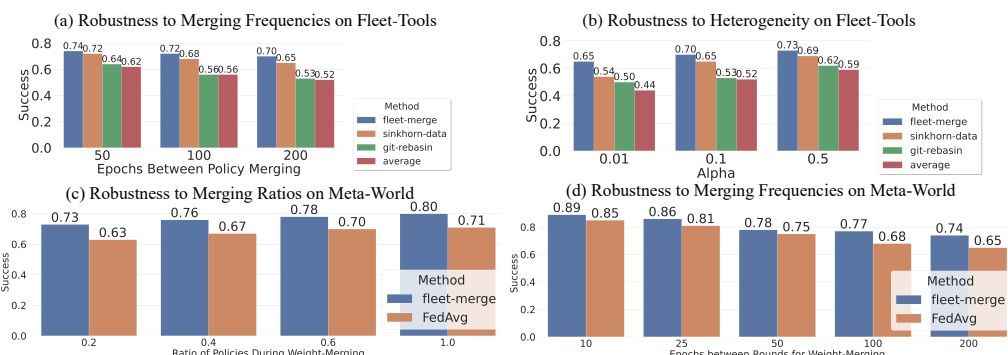

Figure 4: Results on Iterative Merging. We showed that FedAvg with our merged algorithm achieves better performance on both FLEET-TOOLS and Meta-World benchmarks, when changing **(b)** Dirichlet parameters, **(c)** partial participation ratios (25 tasks), and **(a,d)** communication epochs. The performance is upper-bounded by joint training (which pools all the data together for training).

certain multi-task settings. Additionally, because control policies exhibit symmetries to general invertible linear transformations, we find that the gradient-based alignment approach performs the best. This directly contrasts what we find when merging RNN policies with nonlinear activations. Indeed, in the ablations depicted in Figure 12 in Appendix F, we find that failing to project the "soft permutation" matrices back to hard permutations leads to non-trivial degradation in performance when merging a variety of neural network architectures. Thus, our linear experiments demonstrate that the group structure of policy invariances can be essential for proper algorithm design.

## 5.2 ITERATIVE MERGING

We also evaluate the performance of our framework when merging models over the course of training, i.e., *iterative merging*. We focus on multi-task problems in this setting and try to minimize the merging frequencies and ratios under high non-IIDness. Specifically, at every $M$ epoch, we merge models according to different methods, and then reinitialize all the individual models $\theta_i$ at the merged model $\theta_{\mathrm{mrg}}$. This scheme allows more communication rounds between the robots and the designer, trading-off the biases caused by local training when there is a high data heterogeneity across robots. This scheme connects to the popular distributed learning protocol of federated learning (McMahan et al., 2017). See also Section 3.1 and Algorithm 1 for more details of this setting.

**FLEET-TOOLS.** We first evaluate the performance of the full FLEET-MERGE algorithm in FLEET-TOOLS, as plotted in Figure 4 (a)&(b). We find that the performance of our method degrades gradually as the frequency of iterative merging decreases (i.e., the number of epochs between merging increases), which is essential in fleet-learning applications. We also have real-world demos for tool-use using the merged policy from FLEET-MERGE in simulation, which are deferred to Appendix C.2. In Figure 4 (b), we show that FLEET-MERGE also has the best absolute performance and resilience to dataset heterogeneity in the Meta-World benchmark.

**Meta-World.** Figure 4 (c)&(d) study iterative merging in Meta-World. We find that FLEET-MERGE is resilient to a lower frequency of weight merging (which allows more epochs of local training between merges) and to a smaller fraction of models being merged per round (which we refer to as the "participation ratio" of the agents in the fleet). These two together show that communication between agents in the fleet can be reduced without significantly harming the performance of the merged model. We observe similar results across different neural network architectures, different inputs and metrics, and in large-scale settings. We defer the detailed results to Appendix D. Notably, we can learn a multi-task learning policy to solve all 50 manipulation tasks jointly in Meta-World, without pooling the data together.

## 6 CONCLUSION

We studied policy merging, a framework for *fleet learning* of control policies from distributed and potentially heterogeneous datasets, by aggregating the parameters of the (trained) policies. We developed new algorithms to merge multiple policies by taking into account their parameterization ambiguity using recurrent neural networks. Finally, we proposed a novel robotic manipulation benchmark, FLEET-TOOLS, in generalizable tool-use tasks, which might be of broader interest.

## 7 ACKNOWLEDGEMENT

The authors would like to thank many helpful discussions from Eric Cousineau and Hongkai Dai at Toyota Research Institute as well as Tao Pang, Rachel Holladay, and Chanwoo Park at MIT. We thank MIT Supercloud for providing computing cluster resources for running the experiments. This work is supported in part by Amazon Greater Boston Tech Initiative and Amazon PO No. 2D-06310236.

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

CONTENTS

## A    DETAILED RELATED WORK

**Parameter invariance in neural networks and control.**    It has long been known that neural networks are invariant under certain transformations or symmetries, of their weights (Hecht-Nielsen, 1990). In particular, *permutation* symmetries arise because the ordering of hidden neurons in neural networks does not affect their input-output behavior. This has been an important topic of recent studies of neural networks loss landscapes (Brea et al., 2019; Entezari et al., 2021; Tatro et al., 2020; Ainsworth et al., 2022) and of methods that process the weights of other networks  (Deutsch et al., 2019; Navon et al., 2023; Zhou et al., 2023). On the other hand, in the controls literature, it is known that the input-output behavior of a linear dynamical system is invariant under any *similarity* transformations on the system matrices (Åström & Murray, 2021; Åström, 2012). Recent works have further studied the optimization landscape of classical linear controllers (Fazel et al., 2018; Zheng et al., 2021; Umenberger et al., 2022; Hu et al., 2023), especially for *dynamic* controllers that have recurrent latent states (Zheng et al., 2021; Umenberger et al., 2022). In our work, we consider the symmetry of general nonlinear policies parametrized by neural networks, particularly recurrent neural networks, and how to leverage this property to merge multiple policies in robotic fleet learning.

**Model merging & Mode connectivity.**    There have been a variety of approaches to merging neural networks by interpolating their weights. For deep neural networks, this typically requires aligning hidden neurons using techniques similar to those in sensor fusion (Poore, 1994). One line of work uses optimal transport to align the weights of distinct models prior to merging (Singh & Jaggi, 2020; Tan et al., 2022), while related works have investigated removing permutation symmetries to align neurons before interpolation (Tatro et al., 2020; Entezari et al., 2021; Ainsworth et al., 2022; Peña et al., 2022). The connectivity property of the landscape when optimizing linear controllers has also been a resurgent research interest (Feng & Lavaei, 2020; Zheng et al., 2021; Bu et al., 2019). Notably, Bu et al. (2019) shows that the parameters of stabilizing static state-feedback linear controllers form a single connected component, and Zheng et al. (2021) shows that output-feedback dynamical controllers can form at most two connected components. Merging neural networks has also been studied in recent years in the context of finetuning foundation models (Wortsman et al., 2022a;b), federated learning (McMahan et al., 2017; Wang et al., 2020a; Yurochkin et al., 2019; Konečný et al., 2016; Wang et al., 2023), multi-task learning (He et al., 2018; Stoica et al., 2023), and studying linear mode connectivity hypothesis for (feedforward) neural networks (Frankle et al., 2020; Entezari et al., 2021; Ainsworth et al., 2022). Most of these works focused on the feedforward NN architecture. A few works have studied model merging for *recurrent* NNs in the context of federated learning (Hard et al., 2018; McMahan et al., 2018; Wang et al., 2020a). However, they either used *direct averaging* without accounting for the permutation symmetries (Hard et al., 2018; McMahan et al., 2018), or focused on supervised learning tasks (Wang et al., 2020a). In our work, we aim to study how well these insights can be applied to policy learning and control settings, with recurrent neural network parameterization, and awareness of permutation invariance. Compared with the most related work Wang et al. (2020a), we also develop a new merging approach based on "soft" permutations (see Section 3 for a formal introduction) of the neurons, allowing a more efficient implementation and focus on robotic control tasks.

**Multi-task policy learning.**    Multi-task learning and models that exhibit multi-task behaviors have shown impressive successes in computer vision (Zhang & Yang, 2018; Standley et al., 2020) and natural language processing (Radford et al., 2019; Collobert & Weston, 2008; Bubeck et al., 2023). Indeed, multi-task learning (Ruder, 2017; Yu et al., 2020), meta-learning (Vilalta & Drissi, 2002; Nichol et al., 2018; Finn et al., 2017), and few-shot learning (Wang et al., 2020b) have long lines of literature. Despite some promising recent attempts in robotics (Brohan et al., 2022; Shridhar et al., 2023), the dominant paradigm in robotics is still to conduct single-task data collection and training on a single domain. Recently, such multi-task paradigms have also been extended to linear control settings, see e.g., Zhang et al. (2022); Wang et al. (2022a). Our work is also related to the studies on the effects of distribution shifts and diverse data distributions (Agarwal et al., 2021; Koh et al., 2021). Inspired by the increasing scale of robot fleets deployed in the real-world, building a policy learning framework at scale requires us to handle distribution heterogeneity and communication efficiency (Kalashnikov et al., 2021; Herzog et al., 2023; Driess et al., 2023). Also note that the setting we consider, with multiple datasets, include but are not limited to multi-tasks settings, as the datasets may come from different experts in imitation learning, or different time or operating conditions of the

same agent/policy in the same task. While provable guarantees exist for imitation in the single-task setting Block et al. (2023), understanding the implications of behavior cloning in multi-task settings from a theoretical angle is an exciting direction for future work.

# B PERMUTATION INVARIANCE FOR RNNS

## B.1 PROOF OF PROPOSITION 3.1

In this section, we prove Proposition 3.1 in the main paper. Recall that when $\mathcal{P} = (\mathbf{P}^0, \mathbf{P}^1, \ldots, \mathbf{P}^L) \in \mathcal{G}_{\text{perm}}$, it acts on RNN models via

$$(\mathbf{W}_{\text{rec}}^\ell, \mathbf{W}_{\text{ff}}^{\ell-1}, \mathbf{b}^{\ell-1}) \mapsto (\mathbf{P}^\ell \mathbf{W}_{\text{rec}}^\ell (\mathbf{P}^\ell)^\top, \mathbf{P}^\ell \mathbf{W}_{\text{ff}}^{\ell-1} (\mathbf{P}^{\ell-1})^\top, \mathbf{P}^\ell \mathbf{b}^{\ell-1}), \tag{B.1}$$

where we note that we replace the matrix inverse in Eq. (B.1) with matrix transpose because the two coincide for permutation matrices. We now note that the group operation of $\mathcal{P}$ on weights can be decomposed as the computation of applying operators as follows

$$(\mathbf{P}^0, \mathbf{P}^1, \ldots, \mathbf{P}^L) \circ (\theta) = (\mathbf{P}^0, \mathbf{I}, \ldots, \mathbf{I}) \circ (\mathbf{I}, \mathbf{P}^1, \ldots, \mathbf{I}) \circ \cdots \circ (\mathbf{I}, \mathbf{I}, \ldots, \mathbf{P}^L) \circ \theta$$

$$= (\mathbf{I}, \mathbf{P}^1, \ldots, \mathbf{I}) \circ (\mathbf{I}, \mathbf{P}^2, \ldots, \mathbf{I}) \circ \cdots \circ (\mathbf{I}, \mathbf{I}, \ldots, \mathbf{P}^{L-1}) \circ \theta, \tag{B.2}$$

where for the second equality, we recall that $\mathcal{G}_{\text{perm}}$ is defined so that its elements $\mathcal{P} = (\mathbf{P}^0, \mathbf{P}^1, \ldots, \mathbf{P}^L) \in \mathcal{G}_{\text{perm}}$, have $\mathbf{P}^0$ and $\mathbf{P}^L$ equal to the identity (i.e., we do not permute inputs or outputs). It suffices to show that the RNN model is invariant to applying this operation on a specific layer, i.e.,

$$\mathcal{P} = (\mathbf{I}, \mathbf{I}, \ldots, \mathbf{I}, \mathbf{P}^\ell, \mathbf{I}, \ldots, \mathbf{I}). \tag{B.3}$$

We now prove the following claim.

**Claim B.1.** Let $\ell$ be such that $0 < \ell < L$, and let $\mathcal{P}$ be of the form Eq. (B.3), i.e., $\mathbf{P}^\ell$ is the only non-identity. Let $h_t^{\ell'}$ be the hidden state at time $t$ and layer $0 \leq \ell' \leq L$ of model $\theta$, starting at $h_0^{\ell'} = 0$.[3] Let $\tilde{h}_\ell^t$ denote the same, but for the permuted model $\tilde{\theta} = \mathcal{P}(\theta)$. Then, for all $t$,

$$\tilde{h}_t^{\ell'} = \begin{cases} h_t^{\ell'} & \ell' \neq \ell \\ \mathbf{P}^\ell h_t^\ell & \ell' = \ell \end{cases}. \tag{B.4}$$

Note that this claim implies Proposition 3.1 because (B.2) shows that any $\mathcal{P}$ can be decomposed into permutations of the form (B.3) for $0 < \ell < L$. In particular, because $\ell \neq L$, (B.4) means that $h_t^L = \tilde{h}_t^L$, as needed.

*Proof of Claim B.1.* It is clear that $\tilde{h}_t^{\ell'} = h_t^{\ell'}$ for layers $\ell' < \ell$ and $t \geq 0$. It remains to consider layers $\ell' = \ell$ and $\ell' > \ell$.

Let's start with $\ell' = \ell$. We argue by induction on $t$ that $\tilde{h}_t^\ell = \mathbf{P}^\ell h_t^\ell$. Let's start with $t = 0$. By assumption we have that $\tilde{h}_t^\ell = 0 = \mathbf{P}^\ell 0 = \mathbf{P}_t^\ell h_t^\ell$. This proves the base case. For general $t > 0$, by specializing (B.1) to permutation of the form (B.3), we have

$$\tilde{h}_t^\ell = \sigma(\mathbf{P}^\ell \mathbf{W}_{\text{rec}}^\ell (\mathbf{P}^\ell)^\top \tilde{h}_{t-1}^\ell + \mathbf{P}^\ell \mathbf{W}_{\text{ff}}^{\ell-1} \tilde{h}_t^{\ell-1} + \mathbf{P}^\ell \mathbf{b}^{\ell-1})$$

$$= \mathbf{P}^\ell \sigma(\mathbf{W}_{\text{rec}}^\ell (\mathbf{P}^\ell)^\top \tilde{h}_{t-1}^\ell + \mathbf{W}_{\text{ff}}^{\ell-1} \tilde{h}_t^{\ell-1} + \mathbf{b}^{\ell-1})$$

because any entry-wise activation $\sigma(\cdot)$ commutes with $\mathbf{P}^\ell$. As noted above, we have $\tilde{h}_t^{\ell-1} = h_t^{\ell-1}$. Moreover, by the inductive hypothesis, $\tilde{h}_{t-1}^\ell = \mathbf{P}^\ell h_{t-1}^\ell$. Thus, we have

$$\tilde{h}_t^\ell = \mathbf{P}^\ell \sigma(\mathbf{W}_{\text{rec}}^\ell (\mathbf{P}^\ell)^\top \mathbf{P}^\ell h_{t-1}^\ell + \mathbf{W}_{\text{ff}}^{\ell-1} \tilde{h}_t^{\ell-1} + \mathbf{b}^{\ell-1})$$

$$= \mathbf{P}^\ell \sigma(\mathbf{W}_{\text{rec}}^\ell h_{t-1}^\ell + \mathbf{W}_{\text{ff}}^{\ell-1} \tilde{h}_t^{\ell-1} + \mathbf{b}^{\ell-1})$$

$$= \mathbf{P}^\ell h_t^\ell.$$

This proves what we need for $\ell' = \ell$.

---

[3]This condition can be generalized to requiring that $\tilde{h}_0^{\ell'} = \mathbf{P}^\ell h_0^{\ell'}$ for $0 < \ell' < L$ and $\tilde{h}_0^{\ell''} = h_0^{\ell''}$ for $\ell'' \in \{0, L\}$.

For $\ell > \ell'$, it suffices to prove the case when $\ell' = \ell + 1$, because the weights for all subsequent layers are not changed by (B.3) and all layers above $\ell + 1$ depend only on the $\ell + 1$'s input. Again, we induct over $t$. By assumption $h_t^{\ell+1} = 0 = \tilde{h}_t^{\ell+1}$. Moreover, permutations of the form (B.3) mean that

$$\tilde{h}_t^{\ell+1} = \sigma(\mathbf{W}_{\text{rec}}^{\ell+1}\tilde{h}_{t-1}^{\ell+1} + \mathbf{W}_{\text{ff}}^{\ell}(\mathbf{P}^{\ell})^{\top}\tilde{h}_t^{\ell} + \mathbf{b}^{\ell}).$$

Using the inductive hypothesis that $\tilde{h}_{t-1}^{\ell+1} = h_t^{\ell+1}$ and the fact that $\tilde{h}_t^{\ell} = \mathbf{P}^{\ell}h_t^{\ell}$ proven above, we have

$$\tilde{h}_t^{\ell+1} = \sigma(\mathbf{W}_{\text{rec}}^{\ell+1}h_{t-1}^{\ell+1} + \mathbf{W}_{\text{ff}}^{\ell}(\mathbf{P}^{\ell})^{\top}\mathbf{P}^{\ell}h_t^{\ell} + \mathbf{b}^{\ell})$$
$$= \sigma(\mathbf{W}_{\text{rec}}^{\ell+1}h_{t-1}^{\ell+1} + \mathbf{W}_{\text{ff}}^{\ell}h_t^{\ell} + \mathbf{b}^{\ell}) = h_t^{\ell+1},$$

concluding the proof of the claim. $\qquad\square$

## B.2 Invariance for ReLU Activations

For RNNs with common activation functions, we can study a more general group of symmetries beyond just the permutations of Proposition 3.1. Again, consider the action of a general $\mathcal{P} = (\mathbf{I}, \mathbf{P}^1, \mathbf{P}^2, \dots, \mathbf{P}^{L-1}, \mathbf{I}) \in \mathcal{G}_{\text{lin}}$, acting as

$$(\mathbf{W}_{\text{rec}}^{\ell}, \mathbf{W}_{\text{ff}}^{\ell-1}, \mathbf{b}^{\ell-1}) \mapsto \left(\mathbf{P}^{\ell}\mathbf{W}_{\text{rec}}^{\ell}(\mathbf{P}^{\ell})^{-1}, \mathbf{P}^{\ell}\mathbf{W}_{\text{ff}}^{\ell-1}(\mathbf{P}^{\ell-1})^{-1}, \mathbf{P}^{\ell}\mathbf{b}^{\ell-1}\right). \tag{B.5}$$

**Proposition B.1.** Let $\sigma(\cdot)$ be any activation function, and consider $\mathcal{P} \in \mathcal{G}_{\text{lin}}$ acting as above. Suppose that, for each $\ell$, $\mathbf{P}^{\ell}$ commutes with $\sigma$, in the sense that $\mathbf{P}^{\ell}\sigma(\cdot) = \sigma(\mathbf{P}^{\ell}(\cdot))$. Then, the output of the RNN is invariant to the transformation induced by $\mathcal{P}$. In particular, if $\sigma(\cdot) = \text{ReLU}(\cdot)$, then the RNN output is invariant to the set $\mathcal{P} \in \mathcal{G}_{\text{scaled}}$, where $\mathcal{G}_{\text{scaled}} \subset \mathcal{G}_{\text{lin}}$ is the set of transformations such that $\mathbf{P}^{\ell} = \tilde{\mathbf{P}}^{\ell}\mathbf{D}^{\ell}$ are scaled permutations, with $\tilde{\mathbf{P}}^{\ell}$ being a permutation matrix and $\mathbf{D}^{\ell}$ being a diagonal matrix with strictly positive elements.[4]

The proof of Proposition B.1 follows exactly as in the proof of Proposition 3.1 above, by replacing transposes with inverses for the general case. The special case of the ReLU network follows from checking that the ReLU operation commutes with scaled permutations.

Even though RNNs are invariant to scaled permutations of their weights, we follow previous model merging work and focus only on the permutations (i.e., ignoring the scaling) (Ainsworth et al., 2022; Cuturi, 2013). As argued by the linear mode connectivity hypothesis (Entezari et al., 2021), scaling symmetries should not appear in practice because the *implicit regularization* of the training algorithms, e.g., stochastic gradient descent (Neyshabur et al., 2014; Smith et al., 2021), controls the scale of the resulting weights. Here, we formalize the argument by proving that implicit regularization uniquely determines the scaling matrix.

**Proposition B.2.** For an RNN $\theta$, define the norm

$$\|\theta\|_2^2 = \|\mathbf{W}_{\text{ff}}^{L-1}\|_{\text{F}}^2 + \|\mathbf{b}^{L-1}\|^2 + \sum_{0<\ell<L} \|\mathbf{W}_{\text{rec}}^{\ell}\|_{\text{F}}^2 + \|\mathbf{W}_{\text{ff}}^{\ell-1}\|_{\text{F}}^2 + \|\mathbf{b}^{\ell-1}\|^2. \tag{B.6}$$

Let $\theta$ be any model for which every bias term $\mathbf{b}^{\ell}$ has strictly nonzero entries. Define the set

$$\mathcal{X}(\theta) = \underset{\theta', \mathcal{P} \in \mathcal{G}_{\text{scaled}}}{\arg\min} \|\theta'\|_2 \quad \text{s.t. } \theta' = \mathcal{P}(\theta), \tag{B.7}$$

of minimal norm models which correspond to transforming $\theta$ by a scaled permutation. Then, for every $\theta', \theta'' \in \mathcal{X}(\theta)$, there exists a transformation consisting only of (non-scaled) permutations $\mathcal{P} \in \mathcal{G}_{\text{perm}}$ such that $\theta' = \mathcal{P}(\theta'')$.

*Proof.* Note that if $\theta' \in \mathcal{X}(\theta)$, $\mathcal{X}(\theta') = \mathcal{X}(\theta)$ because $\mathcal{G}_{\text{scaled}}$ is a group. Thus, it suffices to prove a slightly simpler claim:

**Claim B.2.** Suppose $\theta \in \mathcal{X}(\theta)$ and $\theta' \in \mathcal{X}(\theta)$, and let $\mathcal{P}$ be such that $\mathcal{P} = (\mathbf{I}, \tilde{\mathbf{P}}^1\mathbf{D}^1, \dots, \tilde{\mathbf{P}}^{L-1}\mathbf{D}^{L-1}, \mathbf{I})$, where $\tilde{\mathbf{P}}$-matrices are permutation matrices and $\mathbf{D}$ matrices are diagonal matrices with positive diagonal elements. Then, we must have

$$\mathbf{D}^{\ell} = \mathbf{I}_{d_{\ell}}, \quad 0 < \ell < L, \tag{B.8}$$

where $d_{\ell}$ is the dimension of the $\mathbf{D}^{\ell}$.

---

[4]Note that this is a group because $\tilde{\mathbf{P}}^{\ell}\mathbf{D}^{\ell} = \tilde{\mathbf{P}}^{\ell}\tilde{\mathbf{D}}^{\ell}$ for some other $\tilde{\mathbf{D}}^{\ell}$.

We now prove the claim. We expand
$$\|\mathcal{P}(\theta)\|_2^2$$
$$= \|\mathbf{W}_{\text{ff}}^{L-1}(\tilde{\mathbf{P}}^{L-1}\mathbf{D}^{L-1})^{-1}\|_{\text{F}}^2 + \|\mathbf{b}^L\|^2$$
$$+ \sum_{0<\ell<L} \|(\tilde{\mathbf{P}}^l\mathbf{D}^l)\mathbf{W}_{\text{rec}}^{\ell}(\mathbf{D}^{\ell})^{-1}(\tilde{\mathbf{P}}^l\mathbf{D}^{\ell})^{-1}\|_{\text{F}}^2 + \|(\tilde{\mathbf{P}}^l\mathbf{D}^{\ell})\mathbf{W}_{\text{ff}}^{\ell-1}(\tilde{\mathbf{P}}^l\mathbf{D}^{\ell-1})^{-1}\|_{\text{F}}^2 + \|(\tilde{\mathbf{P}}^{\ell}\mathbf{D}^l)\mathbf{b}^{\ell-1}\|^2$$
$$= \|\mathbf{W}_{\text{ff}}^{L-1}(\mathbf{D}^{L-1})^{-1}(\tilde{\mathbf{P}}^{L-1})^{-1}\|_{\text{F}}^2 + \|\mathbf{b}^{L-1}\|^2$$
$$+ \sum_{0<\ell<L} \|(\tilde{\mathbf{P}}^l\mathbf{D}^l)\mathbf{W}_{\text{rec}}^{\ell}(\mathbf{D}^{\ell})^{-1}(\tilde{\mathbf{P}}^l)^{-1}\|_{\text{F}}^2 + \|(\tilde{\mathbf{P}}^l\mathbf{D}^{\ell})\mathbf{W}_{\text{ff}}^{\ell-1}(\mathbf{D}^{\ell-1})^{-1}(\tilde{\mathbf{P}}^l)^{-1}\|_{\text{F}}^2 + \|(\tilde{\mathbf{P}}^{\ell}\mathbf{D}^l)\mathbf{b}^{\ell-1}\|^2$$
$$= \|\mathbf{W}_{\text{ff}}^{L-1}(\mathbf{D}^{L-1})^{-1}\|_{\text{F}}^2 + \|\mathbf{b}^{L-1}\|^2$$
$$+ \sum_{0<\ell<L} \|\mathbf{D}^l\mathbf{W}_{\text{rec}}^{\ell}(\mathbf{D}^{\ell})^{-1}\|_{\text{F}}^2 + \|\mathbf{D}^{\ell}\mathbf{W}_{\text{ff}}^{\ell-1}(\mathbf{D}^{\ell-1})^{-1}\|_{\text{F}}^2 + \|\mathbf{D}^l\mathbf{b}^{\ell-1}\|^2$$

$$= \|\mathbf{b}^{L-1}\|^2 + \sum_{i=1}^{d_L}\sum_{j=1}^{d_{L-1}} \mathbf{W}_{\text{ff}}^{L-1}[i,j]^2\mathbf{D}^{L-1}[j,j]^{-2}$$

$$+ \sum_{0<\ell<L}\sum_{i,j=1}^{d_\ell} \mathbf{D}^l[i,i]^2\mathbf{W}_{\text{rec}}^{\ell}[i,j]^2\mathbf{D}^l[j,j]^{-2}$$

$$+ \sum_{0<\ell<L}\sum_{i=1}^{d_\ell}\sum_{j=1}^{d_{\ell-1}} \mathbf{D}^{\ell}[i,i]^2\mathbf{W}_{\text{ff}}^{\ell-1}[i,j]^2\mathbf{D}^{\ell-1}[j,j]^{-2} + \sum_{0<\ell<L}\sum_{i=1}^{d_\ell} \mathbf{D}^l[i,i]^2\mathbf{b}^{\ell-1}[i]^2,$$

where $[i,j]$ indexes matrices and $[i]$ indexes vectors, and where we use that $\mathbf{D}^\ell$ are diagonal matrices. Next, let us represent the diagonals of the matrices $\mathbf{D}^\ell[i,i] = \exp(\tau_{\ell,i})$ as exponentials, which is valid because they are strictly positive. In this representation, we have

$$\|\mathcal{P}(\theta)\|_2^2 = \|\mathbf{b}^{L-1}\|^2 + \sum_{i=1}^{d_L}\sum_{j=1}^{d_{L-1}} \mathbf{W}_{\text{ff}}^{L-1}[i,j]^2 e^{-2\tau_{L-1,j}} + \sum_{0<\ell<L}\sum_{i,j=1}^{d_\ell} \mathbf{W}_{\text{rec}}^{\ell}[i,j]^2 e^{2(\tau_{\ell,i}-\tau_{\ell,j})}$$

$$+ \sum_{0<\ell<L}\sum_{i=1}^{d_\ell}\sum_{j=1}^{d_{\ell-1}} \mathbf{W}_{\text{ff}}^{\ell-1}[i,j]^2 e^{2(\tau_{\ell,i}-\tau_{\ell-1,j})} + \sum_{0<\ell<L}\sum_{i=1}^{d_\ell} e^{2\tau_{\ell,i}}\mathbf{b}^{\ell-1}[i]^2$$

$$:= F(\{\tau_{\ell,i}\}_{1\le i\le d_\ell, 0<\ell<L}).$$

We now observe that the function $F(\cdot)$ is convex in its arguments, because it is the sum over exponentials of linear functions of $\{\tau_{\ell,i}\}_{1\le i\le d_\ell, 0<\ell<L}$ with positive coefficients. In fact, it is *strictly* convex, because all but the last terms are convex, and the term $\sum_{0<\ell<L}\sum_{i=1}^{d_\ell} e^{2\tau_{\ell,i}}\mathbf{b}^{\ell-1}[i]^2$ is strictly convex under our assumption that none of the bias weights are zero.

Therefore, there is a unique $\{\tau_{\ell,i}^\star\}_{1\le i\le d_\ell, 0<\ell<L}$ which minimizes $F(\cdot)$. Thus, by assumption that $\theta' \in \mathcal{X}(\theta)$, we must have that

$$\mathbf{D}^\ell[i,i] = \exp(\tau_{\ell,i}^\star), \quad 1\le i\le d_\ell, 0<\ell<L.$$

But similarly, since $\theta \in \mathcal{X}(\theta)$, then by writing and $\theta = (\mathbf{I},\mathbf{I},\ldots\mathbf{I})\circ\theta$ as the identity transformation applied to itself, we must have that

$$\exp(\tau_{\ell,i}^\star) = (\mathbf{I}_{d_\ell})[i,i] = 1.$$

Thus, we conclude that

$$\mathbf{D}^\ell[i,i] = 1,$$

as needed. $\qquad\square$

## C  FLEET-TOOLS BENCHMARK

### C.1  IMPLEMENTATION DETAILS

In this section, we introduce more details of our newly developed tool-use benchmark for fleet policy learning, based on the Drake simulator (Tedrake & the Drake Development Team, 2019).

**Expert tasks.**  To scale the generation of expert demonstrations for the tool-use tasks, we specify the tasks via key points (Manuelli et al., 2022) when generating the expert trajectories. We mainly consider four skills as examples (or task families): {`wrench`, `hammer`, `spatula`, and `knife`} – consisting of tasks involving the eponymous tool-type. Specifically, we use `use a spanner to apply wrenches`, `use a hammer to hit`, `use a spatula to scoop`, and `use a knife to split`. All of these tasks have a common feature that the skills can be reasonably, certainly not perfectly, solved by specifying a few sparse keypoints. For instance, in the wrench tasks as shown in Figure 5, the wrench needs to reach the nut first and then rotate by 45 degrees while maintaining contact. The same strategy can be applied to a set of nuts and wrenches. We pick these tasks because they provide a balance of common robotic tasks that require precision, dexterity, and generality in object-object affordance.

**Keypoint transformation optimization.**  We denote the end effector frame and world frame as $E$ and $W$, respectively. For each tool, we specify $n$ keypoints, for example, when $n = 3$, we can specify one point on the tool head, one point on the tool tail, and one point on the side. We define a rigid body transformation $\boldsymbol{X} \in \mathrm{SE}(3) \subset \mathbb{R}^{4 \times 4}$, the current locations of the $n$ keypoints on the tool in the world frame $W$ as $^W p^{\texttt{tool}} = \left[^W p^{\texttt{tool},1}, \cdots, {}^W p^{\texttt{tool},n}\right]$, and the poses in the end effector frame as $^E p^{\texttt{tool}} = \left[^E p^{\texttt{tool},1}, \cdots, {}^E p^{\texttt{tool},n}\right]$, respectively. For the object, we specify $m$ keypoints in the world frame as $^W p^{\texttt{obj}} = \left[^W p^{\texttt{obj},1}, \cdots, {}^W p^{\texttt{obj},m}\right]$. Each point $p$ (either $p = {}^E p^{\texttt{tool},i}$ or $p = {}^W p^{\texttt{obj},j}$) lies in $\mathbb{R}^4$, and is expressed in $(x, y, z, 1)$-homogeneous coordinates. Then, one can solve the following optimization problem, referred to as *kPAM-Opt* (where kPAM stands for *KeyPoint Affordance-based Manipulation*) (Manuelli et al., 2022), to find the transformation $\boldsymbol{X}$:

$$\boxed{\text{kPAM-Opt}} \qquad \min_{\boldsymbol{X}} \qquad \left\| \boldsymbol{X}\,{}^E p^{\texttt{tool}} - {}^W p^{\texttt{tool}} \right\|_F^2 \tag{C.1}$$

$$\text{s.t.} \qquad \left\| \boldsymbol{X}\,{}^E p^{\texttt{tool},i} - {}^W p^{\texttt{obj},j} \right\|_2 \leq \epsilon, \ \ \forall i, j \tag{C.2}$$

$$\alpha_{i,j} - \epsilon \leq \beta_{i,j}^\top \boldsymbol{X} \left( {}^E p^{\texttt{tool},i} - {}^E p^{\texttt{tool},j} \right) \leq \alpha_{i,j} + \epsilon, \ \ \forall i, j \tag{C.3}$$

where we choose $i \in \{1, \cdots, n\}$ and $j \in \{1, \cdots, m\}$, and $\beta_{i,j} \in \mathbb{R}^4$ is a vector on the unit sphere with the fourth dimension $\beta_{i,j}(4) = 0$, $\alpha_{i,j}$ is some constant that represents the target angle alignment, and $\epsilon > 0$ is some relaxation level. Constraint Eq. (C.5) restricts the distance between the keypoints on the tool to be close to some target ones on the object to be $\epsilon$-small. For example, when $i = 1$ and $m = 1$, where we use $^E p^{\texttt{tool},1}$ to denote the keypoint corresponds to the head of the tool, then this constraint ensures the head of the tool, e.g., the hammer head, to be close to the object, e.g., the pin. The composition of $(\alpha_{i,j}, \beta_{i,j}, \epsilon)$ in Eq. (C.6) describes some constraints related to the orientations of the tool (e.g., the hammer head needs to be vertical to the world and stay flat), up to some $\epsilon$-relaxation. The overall formulation in Eq. (C.1)-Eq. (C.3) instantiates the general one in Manuelli et al. (2022).

Although the kPAM procedure above can solve for the desired transformation $\boldsymbol{X}$, it can be computationally challenging to recover the joint angle from $\boldsymbol{X}$ via inverse kinematics. Hence, we propose to solve kPAM optimization in the *joint* space, i.e., optimize over the joint angles $q$ that parameterize the transformation through forward kinematics (i.e., `forward-kinematics`$(q)$), and only add constraints to the keypoints in the transformed frame. Specifically, we solve the following joint-space optimization problem:

$$\boxed{\text{Joint-Space-kPAM-Opt}} \qquad \min_{q, \boldsymbol{X}} \qquad \left\| \boldsymbol{X}\,{}^E p^{\texttt{tool}} - {}^W p^{\texttt{tool}} \right\|_F^2 \tag{C.4}$$

$$\text{s.t.} \qquad \left\| \boldsymbol{X}\,{}^E p^{\texttt{tool},i} - {}^W p^{\texttt{obj},j} \right\|_2 \leq \epsilon, \ \ \forall i, j \tag{C.5}$$

$$\alpha_{i,j} - \epsilon \leq \beta_{i,j}^\top \boldsymbol{X} \left( {}^E p^{\texttt{tool},i} - {}^E p^{\texttt{tool},j} \right) \leq \alpha_{i,j} + \epsilon, \ \ \forall i, j \tag{C.6}$$

$$\boldsymbol{X} = \texttt{forward-kinematics}(q), \tag{C.7}$$

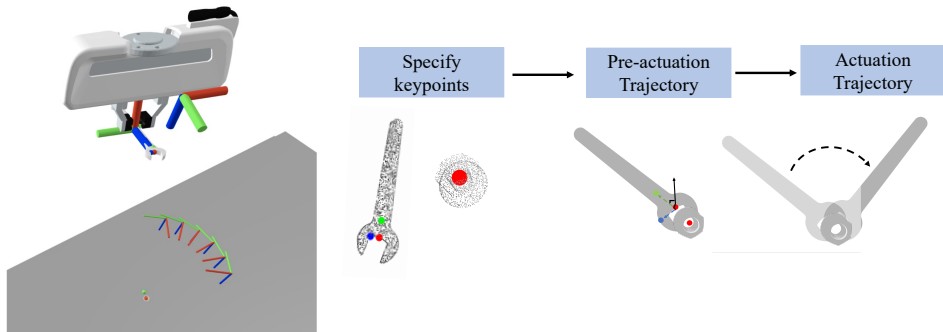

Figure 5: Visualization of the kPAM-based (Manuelli et al., 2022) expert generation pipeline. There are three keypoints defined on the tool and one keypoint defined on the object. The kPAM objective and constraints define an actuation pose for each specific tool-use task. We then optimize for the pre-actuation trajectory and the post-actuation trajectory jointly.

where constraint Eq. (4.4) relates the rigid body transformation $X$ and the output of `forward-kinematics`$(q)$ under a joint angle $q$, to better contrast the formulation in Eq. (C.1)-Eq. (C.3).

We create the optimization problem Eq. (C.4)-Eq. (C.6) in Drake, which is then solved using nonlinear programming solvers as SNOPT (Gill et al., 2005). The intuition of this joint-space optimization is to leverage the multiple solutions in the manipulation tasks to simplify the constraint satisfaction problem in inverse kinematics. For instance, when solving the `wrench` tasks, the entire rotation space around the $z$ axis in the world frame is free, so solving for joint angle directly on these constraints defined on keypoints can be easier than solving the pose-space KPAM problem and then solve inverse kinematics.

**Pre-/Post-actuation optimization.** After finding the joint angle solution $q^\star$, we can work out the pre-actuation and the post-actuation poses (Figure 5). For instance, for `wrench` tasks, there is a standoff pose that is a few centimeters in the negative $z$ axis of the end effector, which are treated as keyframes. Once we construct the trajectory of pose keyframes in the pre-actuation and post-actuation trajectories, we can solve the joint-space trajectory as an optimization problem that further improves the smoothness (by minimizing the norm of joint difference with fixed endpoints), while satisfying the keyframe pose constraints.

Specifically, let $\xi = [\xi[1], \xi[2], ..., \xi[T]]$ denote a joint-space trajectory of length $T$, including trajectory before actuation at some timestep $t^\star < T$, and the post-actuation trajectory from $t^\star + 1$ until $T$. Choose $m \leq T$ indices $\{I_1, \cdots, I_m\} \subseteq \{1, 2, \cdots, T\}$, and specify the corresponding pose keyframe trajectory as $[p_{I_1}, p_{I_2}, ..., p_{I_m}]$. Let $q_1$ be the current joint position, and $q^\star$ be the solution to the previous joint-space KPAM problem Eq. (C.4)-Eq. (C.7). We then solve the following optimization problem:

$$\min_{\xi} \quad \sum_{t=1}^{T-1} \|\xi[t+1] - \xi[t]\|_2^2 \tag{C.8}$$

$$\text{s.t.} \quad \xi[1] = q_1, \qquad \xi[t^\star] = q^\star \tag{C.9}$$

$$\texttt{forward-kinematics}(\xi_{I_k}) = p_{I_k}, \ \forall\, k = 1, 2, \cdots, m, \tag{C.10}$$

which is also solved by nonlinear solvers as SNOPT (Gill et al., 2005) in Drake. This final solution gives a planned trajectory $\xi^\star$ and we can then track this trajectory with low-level controllers to provide demonstrations.

**Discussions.** Our tool-use benchmark and the pipeline above provide a natural and easy way to create (new) tasks in Drake, with only a configuration file of the actuation pose optimization and a few keypoints labels that can be easily obtained via a 3D visualizer. The overall pipeline, which is deterministic by design, is also general enough to act as the expert in the imitation learning settings to represent "category-level" composition and generalization. The convenience of category-level generalization can be an important application domain to study *out-of-support* distribution shift,

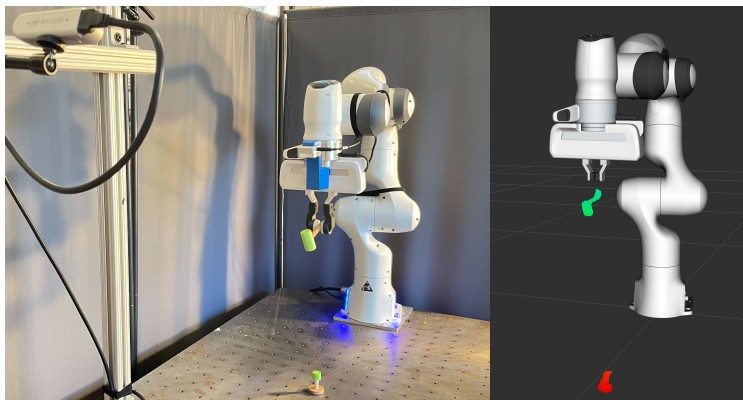

Figure 6: The real-world setup that the benchmark recreates in the simulation with segmented point cloud visualizations.

which can be formulated as combinatorial extrapolation/generalization problems, i.e., generalizing to the unseen tool-object (or tool-task) combinations at test time, despite only being able to see specific tools and objects separately (while not such joint combinations) at training time. See a more formal treatment in Simchowitz et al. (2023).

There are certain limitations of our pipeline. For example, it creates only hand-scripted experts, and there is no feedback in the execution loop, which can be necessary for some dynamic and contact-rich motions (Gao & Tedrake, 2021). Moreover, tool-object and tool-hand contact dynamics modeled in the simulation but are not considered in the current expert formulation. These are all important directions we will address in the future. We believe tool-use tasks, which often involve rich force and contact interactions (?Holladay et al., 2019; Toussaint, 2009), represent one of the key challenging tasks in dexterous manipulation.

## C.2 ENVIRONMENT SETUP AND IMPLEMENTATION DETAILS

We set up two cameras to mimic the real-world setting of Panda robotic arms (see Figure 6). One overhead camera is mounted on the table and the other camera is mounted on the wrist. We have 4 tool-object pairs for each tool in the benchmark (more assets available). At each scene initialization, we randomize the camera extrinsics, robot initial joints, object frictions, mass, and the tool-in-hand pose. We also randomize object texture when using colors and support offline blender rendering. We weld the tool to the robot end effector which forms a kinematic chain. We assume RGB-D images with masks as inputs to the robots. Specifically, we can use this information to fuse pointclouds (Seita et al., 2023; Wang et al., 2022b) for both the tools and the objects, which we resample to contain 1024 points. We also measure joint torques and end effector wrench in the environment. We use the bounded relative end effector motion as the action output. Notably, in the simulation we have a perfect dynamics model that can potentially improve the contact-rich motions in tool-use, which enables us to use a customized operation-space controller (OSC) (Khatib, 1987) with tuned gains to achieve high success rates in solving the tasks. We use hydroelastic contact model with SAP contact solver (Masterjohn et al., 2022) in drake. The environment timesteps are 6Hz and the simulation dynamics step is 250Hz. We use the Ray library to support multiple rollouts in parallel.

We combine the segmented point cloud from a wrist camera and an overhead camera into the tool and object pointcloud, and feed them as observations to the policy. We train the policy using Adam optimizer (Kingma & Ba, 2014) with 200 epochs. The dataset contains 50000 data points for each tool instance, and the batch size is 512. We use 5 users across the merging experiments. Each test trial has 30 different scenes and we take average over 10 trials.

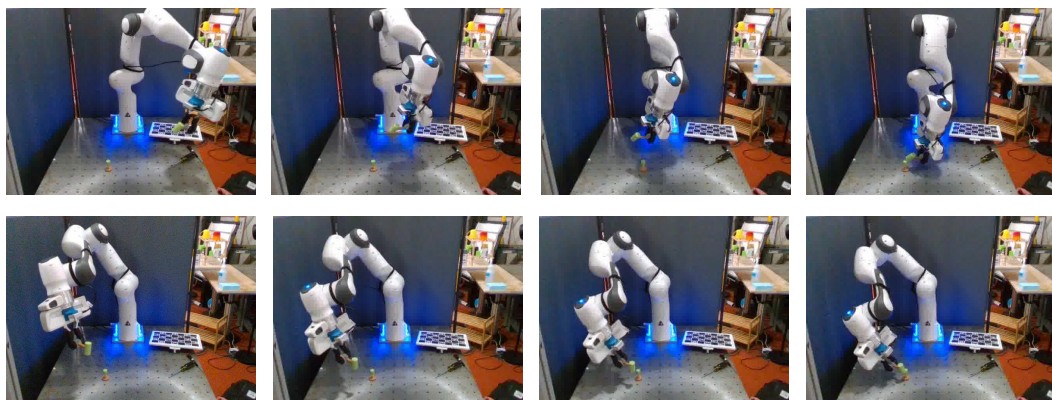

Figure 7: We deployed the merged policies in the real world for the hammering task.

## D    META-WORLD EXPERIMENTS

### D.1    FURTHER EXPERIMENTAL DETAILS

In this section, we provide the experiment details on the Meta-World benchmark (Yu et al., 2020).

**Implementation.**    For state-based input, we use a 4-layer network architecture with 512 hidden dimensions, including multilayer perceptron (MLP), RNN, and Transformer, to parameterize the policies. For image-based input, we use ResNet18 (He et al., 2016) as the default vision encoder and a similar 4-layer MLP for the policy head. We use Adam optimizer with a learning rate of $1e-3$ and batch size of 256. We use 3, 5, and 10 users respectively for the single-shot merging, merging while training, and merging with participation ratio experiments in Figure 10. For validating mode connectivity, we use the same dataset for training two models, but we find that training on different datasets (generated by the same expert) will work as well.

**Evaluation across tasks.**    We evaluate and compare the performance of our methods on large-scale experiments, notably, in the MT-50 tasks in Meta-World. In Figure 10, we provide the comparison of our policy merging methods with two baselines: *joint multi-task training* and *task-specific training*. In particular, in joint training, we pool the data from multiple tasks to train a task-conditioned policy; in task-specific training, we train a policy for each task separately. As shown in Figure 10, in terms of success rates, joint training, and task-specific training can achieve similar performance. We also observed that the more data is used, the gap between the two is smaller. More importantly, it is shown that our policy merging methods can perform on par with these two baselines, by outputting only one single policy (instead of in task-specific training), while without sharing the training data (instead of in joint multi-task training).

**Evaluation across network architectures.**    We also evaluate our policy-merging methods and corresponding baselines across different neural network architectures. In Figure 9, we observe similar performance connectivity behaviors across network architectures, including MLP, RNN, and Transformers. This indicates that our policy merging formulation and algorithms can be general, and not specifically tied to the neural network architectures.

**Norm variations.**    Since RELU neural network's weights are also symmetric up to scaling in between layers, it's a common question to ask whether we need to explicitly account for that. In Figure 8, we show that the network's weights of RNN policies and neural net models at each layer are relatively stable, across different runs on training on the same dataset. The y-axis shows the relative ratio of the weight different norm compared to the weight norm, which is less than $1\%$ in general.

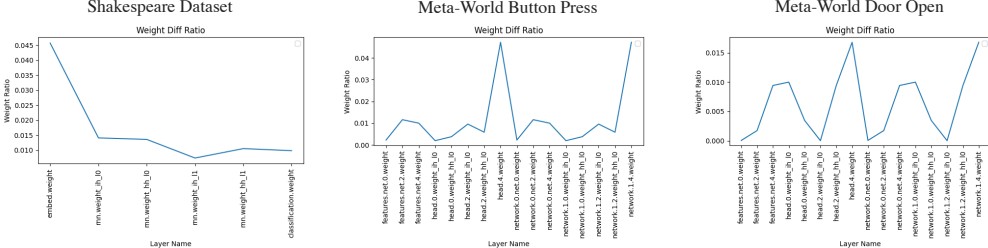

Figure 8: The weight norm ratio (weight difference between runs, compared to weight norms of the first run) of trained RNNs on multiple benchmarks.

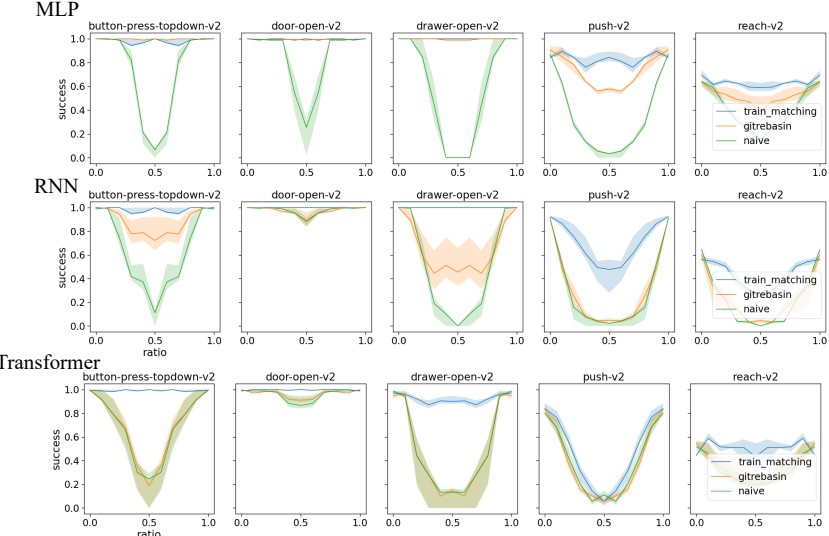

Figure 9: Performance connectivity across architectures. Git rebasin and train matching work consistently across many tasks for every architecture.

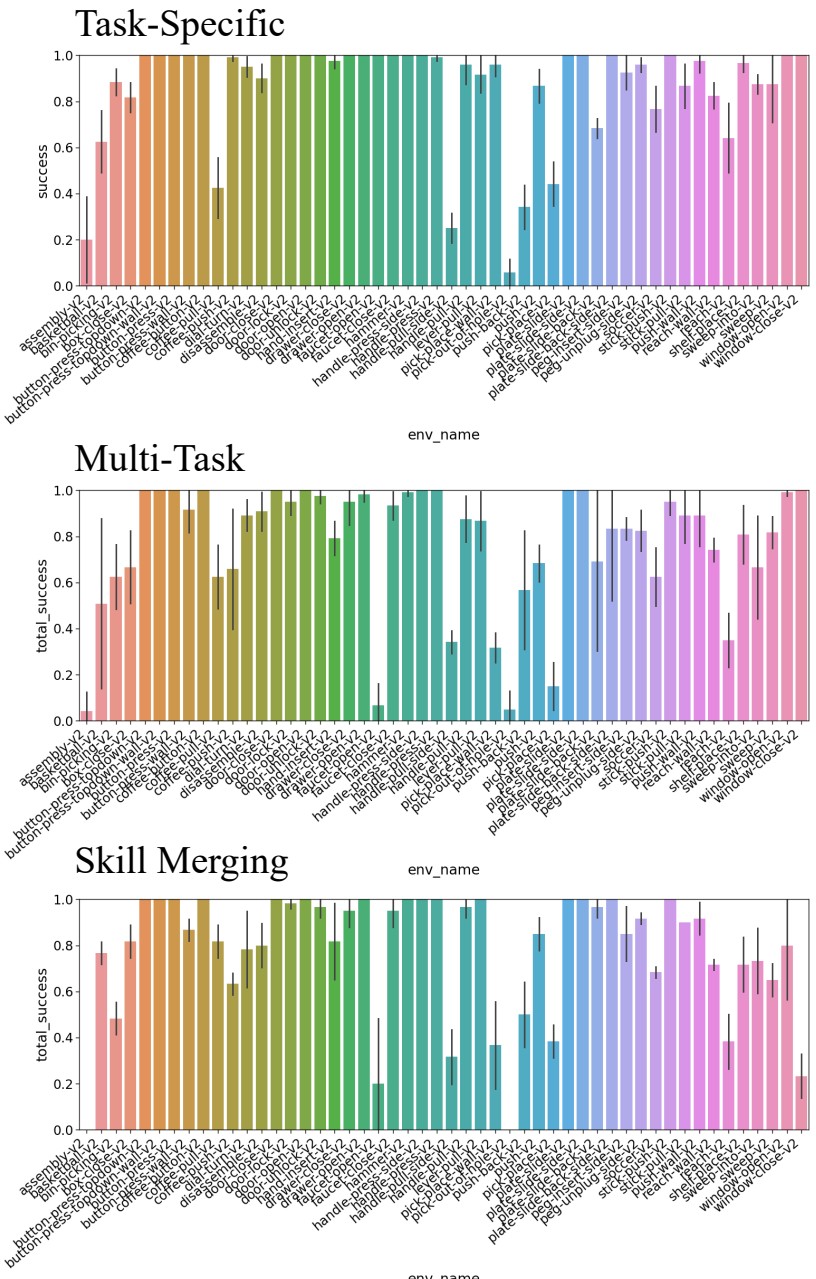

Figure 10: Comparison of joint multi-task training, task-specific training, and policy merging on 50 tasks in Meta-World (MT-50).

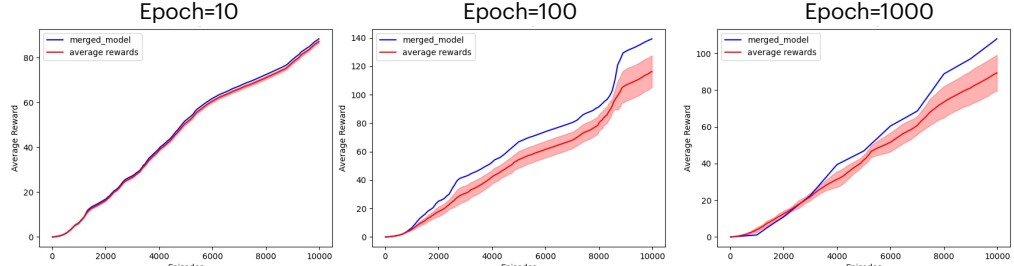

Figure 11: We show that fleet learning can also apply to online reinforcement learning settings where agents continue to gather new data for training. The performance is robust to the communication epochs ranging from 10 to 1000 trajectories between every averaging. The merged model also outperforms the average of each individual model in this case.

# E    ONLINE REINFORCEMENT LEARNING ABLATION

To apply fleet learning to online reinforcement learning, we use the REINFORCE (Sutton et al., 1999) algorithm in the CartPole environment in OpenAI Gym as a proof of concept. In this experiment, we train 5 different policies on 5 instances of the CartPole environment with different random seeds, and average the policies after a certain number of episodes. The experiment (Figure 11) shows that simple averaging can be used when each agent collects more data online and uploads model weights to compose a policy. Note that no data is shared during the learning process.

# F    ALGORITHM ABLATION ON SUPERVISED LEARNING TASKS

In this section, we ablate on different components of our proposed methods of merging multiple models in the one-shot setting, i.e., we only merge the models once, on benchmark supervised learning tasks with different neural network architectures. Specifically, we split the MNIST dataset into $N$ local datasets, with Dirichlet parameters $\alpha$ to create data non-IIDness (see **??** for more details), and we use $L$-layer MLPs to parameterize the models. We lay out the methods we ablate on as follows:

1. **average:** This method naively averages all the weights of the trained model on each local dataset and then tests on the MNIST test set.

2. **git-rebasin:** This method uses the MergeMany algorithm in Ainsworth et al. (2022) to merge all the locally trained models. Specifically, the MergeMany algorithm is an alternation-based procedure: at each round, it first randomly samples one model, and then aligns it with the average of the rest of the models, by using coordinate descent on the aligning loss; the procedure continues until convergence.

3. **sinkhorn-data:** This method similarly applies the alternation-based algorithm above, and replaces the alignment step from Ainsworth et al. (2022) to sinkhorn-rebasin in Peña et al. (2022). Specifically, we use the validation dataset for the alignment. The difference between this alternation-based method and our FLEET-MERGE algorithm (Algorithm 1) is whether we reinitialize the permutation with the hard permutation matrix at each round or not.

4. **FLEET-MERGE (ours):** This is our main algorithm FLEET-MERGE (Algorithm 1) where we use a FedAvg style algorithm to update the permutation parameters in sinkhorn rebasin.

5. **fleet-merge-soft:** This is an ablation method on our algorithm where we replace the hard permutations at line 6 in Algorithm 1 by soft permutation matrices that are computationally more tractable.

6. **fleet-merge-simplex:** This is an ablation method on our algorithm where instead of trying to interpolate between each local model and the averaged model, we randomly sample in the convex hull of $N$ local models and compute the gradients.

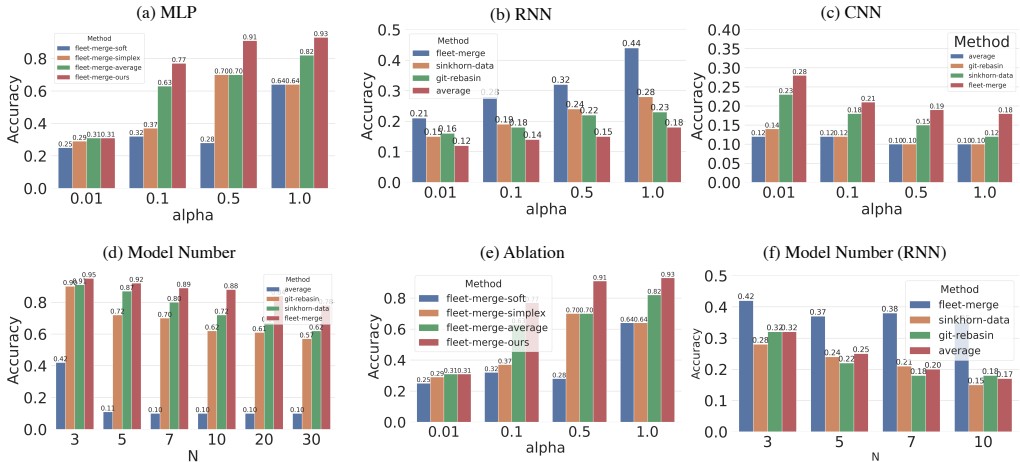

Figure 12: Algorithm Ablation Study on MNIST. Each model owns a shard of the total dataset and our Mani-Rebasin algorithm outperforms all baseline methods.

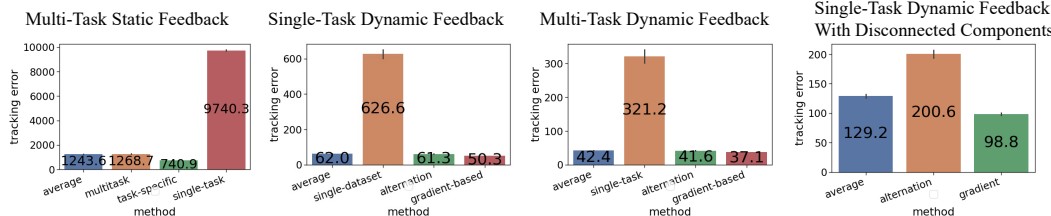

Figure 13: Linear Single Task Results. **(Left)** We observe that even averaging linear policies can often improve performance to almost match joint training on the pooled data for single-task settings, and match task-specific training for multi-task settings. **(Right)** We observe that gradient-based methods to find general invertible matrices perform better than finding permutation matrices with alternations.

7. **fleet-merge-average:** This is an ablation method on our algorithm where instead of trying to interpolate between each local model and the averaged model, we use the average of these $N$ local models and compute the gradients.

As we show in the figure, our methods outperform all the baselines across various neural network architectures, such as MLP on MNIST dataset, RNN on Shakespear dataset from LEAF (Caldas et al., 2018) (sub-figure f), as well as CNN on CIFAR-10 dataset (Krizhevsky et al., 2009) (sub-figure c). We also ablate against the design choices in the algorithm (sub-figure e) and ablate against both non-IIDness (sub-figure a,b,c) and model numbers in the multiple model merging setting (sub-figure d, f).

# G LINEAR POLICY MERGING

## G.1 PROBLEM SETUP

**Linear quadratic control.** A classical sub-setting of the policy setup setting described in Section 2, and one whose optimal policy is recurrent, is that of linear quadratic Gaussian (LQG) control (Åström & Murray, 2021). To respect the notational convention in control theory, we use $(\mathbf{x}_t, \mathbf{y}_t, \mathbf{u}_t)$ to denote the (state, observation, action) tuple of $(s_t, o_t, a_t)$ in Section 2. Let $n$ be the dimension of the state,

$p$ the dimension of the observations, and $m$ the dimension of the control input; that is, $\mathbf{x}_t \in \mathbb{R}^n$, $\mathbf{y}_t \in \mathbb{R}^p$ and $\mathbf{u}_t \in \mathbb{R}^m$. The LQG problem is defined by a linear time-invariant (LTI) dynamical system parameterized by dynamical matrices $(\mathbf{A}, \mathbf{B}, \mathbf{C})$ of appropriate dimenion, evolving according to the following dynamics:

$$\begin{aligned} \mathbf{x}_{t+1} &= \mathbf{A}\mathbf{x}_t + \mathbf{B}\mathbf{u}_t + \mathbf{w}_t \\ \mathbf{y}_t &= \mathbf{C}\mathbf{x}_t + \mathbf{v}_t. \end{aligned} \tag{G.1}$$

Above, for $t \geq 0$, $\mathbf{w}_t$ and $\mathbf{v}_t$ are noise vectors drawn i.i.d. Gaussian from $\mathcal{N}(0, \boldsymbol{\Sigma}_w)$ and $\mathcal{N}(0, \boldsymbol{\Sigma}_v)$ respectively. The initial state $\mathbf{x}_0$ is also drawn from a Gaussian distribution $\mathcal{N}(0, \boldsymbol{\Sigma}_0)$. The classical LQR cost is a positive definite quadratic cost

$$c(\mathbf{x}, \mathbf{u}) := \mathbf{x}^\top \mathbf{Q}\mathbf{x} + \mathbf{u}^\top \mathbf{R}\mathbf{u}, \quad \mathbf{Q} \succ 0, \mathbf{R} \succ 0.$$

The goal of LQG is to minimize the following long-term average cost over policies $\pi$ : $(\mathbf{x}_{1:t}, \mathbf{u}_{1:t-1}) \to \mathbf{u}_t$:

$$\pi^\star \in \operatorname*{argmin}_\pi J(\pi), \quad J(\pi) := \limsup_{T \to \infty} \frac{1}{T}\mathbb{E}^\pi \left[ \sum_{t=0}^{T-1} \mathbf{x}_t^\top \mathbf{Q}\mathbf{x}_t + \mathbf{u}_t^\top \mathbf{R}\mathbf{u}_t \right], \tag{G.2}$$

where above $\mathbb{E}^\pi$ denotes expectation under the closed loop dynamics induced by $\pi$ and the dynamics Eq. (G.1) In the special case with full state observability, we have $\mathbf{C} = I$ and $\boldsymbol{\Sigma}_v = 0$, and the problem is also referred to as linear quadratic regulator (LQR) control.

**Policy parameterization.** It is now classical that under standard controllability and observability assumptions[5], the optimal $\pi^\star$ in Eq. (G.2) is dynamic linear policy, i.e. the specialization of a recurrent policy to this linear setting.

On finite horizons, the optimal control law roughly correspond takes the form $\mathbf{u}_t = \mathbf{K}_{\star,t}\hat{\mathbf{x}}_t$, where the time-varying gain matrices $\mathbf{K}_{\star,t}$ solves a finite-time LQR problem, and where $\hat{\mathbf{x}}_t = \mathbb{E}[\mathbf{x}_t \mid h_t]$, where $h_t$ is the observations $(\mathbf{y}_{1:t}, \mathbf{u}_{1:t-1})$ up to time $t$. Passing to the infinite horizon, the optimal policy takes a form we now describe.

First, we compute an optimal control gain matrix $\mathbf{K}_\star$ can be computed as

$$\mathbf{K}_\star = -(\mathbf{B}^\top \mathbf{P}_\star \mathbf{B} + \mathbf{R})^{-1}\mathbf{B}^\top \mathbf{P}_\star \mathbf{A} \tag{G.3}$$

where $\mathbf{P}_\star$ solves the discrete algebraic Riccati equation:

$$\mathbf{P} = \mathbf{A}^\top \mathbf{P}\mathbf{A} + \mathbf{A}^\top \mathbf{P}\mathbf{B}(\mathbf{B}^\top \mathbf{P}\mathbf{B} + \mathbf{R})^{-1}\mathbf{B}^\top \mathbf{P}\mathbf{A} + \mathbf{Q}. \tag{G.4}$$

Similarly, we compute matrices $\mathbf{L}_\star$ and $\boldsymbol{\Sigma}_\star$ are solve Ricatti equations dual to those for which $\mathbf{K}_\star$ and $\mathbf{P}_\star$ are the solutions:

$$\begin{aligned} \mathbf{L}_\star &= \boldsymbol{\Sigma}_\star \mathbf{C}^\top (\mathbf{C}\boldsymbol{\Sigma}_\star \mathbf{C}^\top + \boldsymbol{\Sigma}_v)^{-1} \\ \boldsymbol{\Sigma}_\star &= \mathbf{A}\boldsymbol{\Sigma}_\star \mathbf{A}^\top - \mathbf{A}\boldsymbol{\Sigma}_\star \mathbf{C}^\top (\mathbf{C}\boldsymbol{\Sigma}_\star \mathbf{C}^\top + \boldsymbol{\Sigma}_v)^{-1}\mathbf{C}\boldsymbol{\Sigma}_\star \mathbf{A}^\top + \boldsymbol{\Sigma}_w. \end{aligned} \tag{G.5}$$

Because of the similarities between the equations defining $\mathbf{L}_\star$ and $\mathbf{K}_\star$, $\mathbf{L}_\star$ is usually referred to as the Kalman gain. Under the optimal policy, the latter is used to define an optimal reccurent estimate $\hat{\mathbf{x}}_t$ of the true state $\boldsymbol{x}_t$, obtained via the Kalman filter equation:

$$\hat{\mathbf{x}}_t = \mathbf{A}\hat{\mathbf{x}}_{t-1} + \mathbf{B}\mathbf{u}_{t-1} + \mathbf{L}_\star(\mathbf{y}_t - \mathbf{C}(\mathbf{A}\hat{\mathbf{x}}_{t-1} + \mathbf{B}\mathbf{u}_{t-1})). \tag{G.6}$$

The optimal control policy can then be expressed as

$$\begin{aligned} \hat{\mathbf{x}}_t &= \mathbf{A}_{\theta^\star}\hat{\mathbf{x}}_{t-1} + \mathbf{B}_{\theta^\star}\boldsymbol{y}_t, \quad \mathbf{u}_t = \mathbf{K}_\star\hat{\mathbf{x}}_t =: \mathbf{C}_{\theta^\star}\hat{\mathbf{x}}_t, \\ \mathbf{A}_{\theta^\star} &:= (\mathbf{A} + \mathbf{B}\mathbf{K}_\star - \mathbf{L}_\star\mathbf{C}(\mathbf{A} + \mathbf{B}\mathbf{K}_\star)), \quad \mathbf{B}_{\theta^\star} := \mathbf{L}_\star, \mathbf{C}_{\theta^\star} := \mathbf{K}_\star \end{aligned} \tag{G.7}$$

which itself is a linear dynamical system, with inputs $(\mathbf{y}_t)_{t \geq 1}$, states $(\hat{\mathbf{x}}_t)_{t \geq 1}$, and outputs $(\mathbf{u}_t)_{t \geq 1}$; i.e. inputs of the control policy are outputs of the system, and vice versa.

This motivates considering a policy class of policies parametrized by $\theta := (\mathbf{A}_\theta, \mathbf{B}_\theta, \mathbf{C}_\theta)$ where $\mathbf{A}_\theta \in \mathbb{R}^{n \times n}$ (i.e., the same dimensions as $\mathbf{A}$), $\mathbf{C}_\theta$ has the same dimension as $\mathbf{B}^\top$, and $\mathbf{B}_\theta$ the same dimensions as $\mathbf{C}^\top$. Following the formula in Eq. (G.7):

$$\hat{\mathbf{x}}_t = \mathbf{A}_\theta\hat{\mathbf{x}}_{t-1} + \mathbf{B}_\theta\mathbf{y}_t, \qquad \mathbf{u}_t = \mathbf{C}_\theta\hat{\mathbf{x}}_t, \tag{G.8}$$

and we can regard $\theta^\star := (\mathbf{A}_{\theta^\star}, \mathbf{B}_{\theta^\star}, \mathbf{C}_{\theta^\star})$ as the optimum of such a set of parameters. Thus, in our behavior cloning objective, we optimize over $\theta$ of the form (G.8). Again, we stressed can be viewed

---

[5]More generally, under stabilizability and detectability assumptions.

as a special case as the RNN parameterization introduced in Section 2, with one hidden layer and no activation functions. We let $\pi_\theta$ denote the corresponding policy, i.e., $J(\pi_\theta)$ is the cost associated with executing the policy induced by $\theta$.

**Feedforward controller.** In the special case where $\mathbf{C} = \mathbf{I}$ and $\boldsymbol{\Sigma}_v = \mathbf{0}$, $\mathbf{y}_t = \mathbf{x}_t$ optimal controller because the *static* (i.e., one-layer feedforward) control law $\mathbf{u}_t = \mathbf{K}_\star \mathbf{x}_t$, where $\mathbf{K}_\star$ is exactly as in (G.3) above. This corresponds to the case of $(\mathbf{A}_\theta, \mathbf{B}_\theta, \mathbf{C}_\theta) = (\mathbf{0}, \mathbf{I}_n, \mathbf{K}_\star)$. Notice that only with $\mathbf{C} = \mathbf{I}$ (but $\boldsymbol{\Sigma}_v \neq \mathbf{0}$), feedforward policies are no longer optimal, because a better estimate $\hat{\mathbf{x}}_t$ of the system state $\boldsymbol{x}_t$ can be obtained by taking the history into account. Nevertheless, we can still view the policy $\mathbf{x}_t \mapsto \mathbf{K}_\star \mathbf{u}_t$ as an *imperfect expert* that one would like to imitate, as studied by Zhang et al. (2022).

## G.2 SYMMETRIES IN NONLINEAR CONTROL

It is both well-known and straightforward that linear control policies are invariant under invertible linear transformations acting as follows

$$(\mathbf{A}_\theta, \mathbf{B}_\theta, \mathbf{C}_\theta) \mapsto (\mathbf{T}\mathbf{A}_\theta\mathbf{T}^{-1}, \mathbf{T}\mathbf{B}_\theta, \mathbf{C}_\theta\mathbf{T}^{-1}) : \det(\mathbf{T}) \neq 0. \tag{G.9}$$

As an example, the policies $(\mathbf{A}_\theta, \mathbf{B}_\theta, \mathbf{C}_\theta)$ and $(\mathbf{A}_\theta, -\mathbf{B}_\theta, -\mathbf{C}_\theta)$ are equivalent, maintaining an internal state which are sign-flips of the other. In the special case of imitating a feedforward expert, there are *no symmetries*. Indeed, if $\mathbf{Kx} = \mathbf{K'x}$ for all $\mathbf{x}$, then $\mathbf{K} = \mathbf{K'}$.

## G.3 DETAILS FOR TRAINING IMITATION LEARNING

In this section, we describe the details for training the imitation learning policy, given in Algorithm 2 below. We begin with the training of dynamic policies. We assume access to a dataset of expert trajectories $\boldsymbol{\tau}^{(i)}$. An important point is initialization. We initialize $\mathbf{A}_\theta = 0$ which ensures that the dynamics defining $\hat{\mathbf{x}}$ are stable We also ensure that $\mathbf{B}_\theta, \mathbf{C}_\theta$ have unit Gaussian entries. Note that $\mathbf{A}_\theta = 0$ is *not* necessarily a critical point of the loss. Indeed, when because when $\mathbf{A}_\theta = 0$, $\hat{\mathbf{x}}_t = \mathbf{A}_\theta \hat{\mathbf{x}}_{t-1} + \mathbf{B}_\theta \mathbf{y}_t = \mathbf{B}_\theta \mathbf{y}_t$, so $\hat{\mathbf{x}}_t$ does not vanish, that thus gradients with respect to $\mathbf{A}_\theta$ can be incorporated. Note that Line 10 in the update requires differentiating through the *entire* dynamics. Notice that while LQG policy optimization is still unsolved, gradient descent for imitation learning is dual to policy search over Kalman filters, and can be solved in a model-free manner due to Umenberger et al. (2022). Feedforward imitation learning is achieved by direct least squares Algorithm 3.

---

**Algorithm 2** Dynamic Linear Policy Learning

1: **Input**: Expert dataset $\mathcal{D} = \{\boldsymbol{\tau}^{(i)} = (\mathbf{y}_1^{(i)}, \mathbf{u}_1^{(i)}, ..., \mathbf{y}_{T-1}^{(i)}, \mathbf{u}_{T-1}^{(i)}, \mathbf{y}_T^{(i)}, \mathbf{u}_T^{(i)})\}$
2: **Initialize** $\mathbf{A}_\theta \leftarrow \mathbf{0}$, and $\mathbf{B}_\theta, \mathbf{C}_\theta$ to have unit Gaussian entries
3: **for** iter $= 1, ..., \text{iter}_{\max}$ **do**
4:     Sample one trajectory $\boldsymbol{\tau} = (\mathbf{y}_1, \mathbf{u}_1, ..., \mathbf{y}_{T-1}, \mathbf{u}_{T-1}, \mathbf{y}_T, \mathbf{u}_T)$
5:     Initiate: $\hat{\mathbf{x}}_1 \leftarrow 0, \ell \leftarrow 0$
6:     **for** $t \in \{1, ..., T\}$ **do**
7:         Update step: $\hat{\mathbf{x}}_t \leftarrow \mathbf{A}_\theta \hat{\mathbf{x}}_{t-1} + \mathbf{B}_\theta \mathbf{y}_t$
8:         Compute control: $\hat{\mathbf{u}}_t \leftarrow \mathbf{C}_\theta \hat{\mathbf{x}}_t$
9:         Update loss: $\ell \leftarrow \ell + \|\mathbf{u}_t - \hat{\mathbf{u}}_t\|_2^2$
10:     Gradient Update: $\theta \leftarrow \theta - \eta \frac{d\ell}{d\theta}$, where gradients are taking through the entire dynamics.
11: **Return**: $\theta = (\mathbf{A}_\theta, \mathbf{B}_\theta, \mathbf{C}_\theta)$

---

**Algorithm 3** Static Linear Policy Learning

1: **Input**: Expert dataset $\mathcal{D} = \{(\mathbf{u}^{(i)}, \mathbf{y}^{(i)})\}$
2: Solve $\hat{\mathbf{K}} \in \operatorname{argmin}_{\mathbf{K}} \sum_i \|\mathbf{y}^{(i)} - \mathbf{K}\mathbf{u}^{(i)}\|^2$ by least squares
3: **Return**: $\hat{\mathbf{K}}$

---

## G.4 Methods for Linear Policy Merging

In the linear setting, we propose two different types of policy merging algorithms to combine the policies. Both algorithms *alternate* between updating each model's transformation matrix and the merged policy parameters, in order to avoid the problem caused by the nonconvexity of the loss. The first algorithm leverages closed-form solutions such as linear assignments (see this subsection, Appendix G.4.1), whereas the second algorithm is based on gradient-descent updates (see Appendix G.4.2).

### G.4.1 Method 1: Permutation-Based Alignment

Our first approach is to focus on finding *permutation transformations* to merge the models, despite that for linear dynamic policies, they are invariant to general invertible transformation matrices. This type of approach has also been used in the nonlinear setting, as we introduced in Section 3.1. Specifically, we propose to minimize the sum of the distances of each individual model $\theta_i = (\mathbf{A}_{\theta_i}, \mathbf{B}_{\theta_i}, \mathbf{C}_{\theta_i})$ to the merged model $\bar{\theta} = (\mathbf{A}_{\bar{\theta}}, \mathbf{B}_{\bar{\theta}}, \mathbf{C}_{\bar{\theta}})$ through permutations $\{\mathbf{P}_i\}_{i\in[N]}$[6]:

$$\min_{\bar{\theta},\{\mathbf{P}_i\}_{i=1}^N \in \mathcal{G}_{\text{perm}}^N} \quad g\left(\bar{\theta}, \{\mathbf{P}_i\}_{i=1}^N\right) := \sum_{i=1}^N \left\|\mathbf{A}_{\bar{\theta}} - \mathbf{P}_i^\top \mathbf{A}_{\theta_i}\mathbf{P}_i\right\|_F^2 + \left\|\mathbf{B}_{\bar{\theta}} - \mathbf{P}_i^\top \mathbf{B}_{\theta_i}\right\|_F^2 + \left\|\mathbf{C}_{\bar{\theta}} - \mathbf{C}_{\theta_i}\mathbf{P}_i\right\|_F^2.$$

(G.10)

Despite the loss in Eq. (G.10) being jointly nonconvex in the variables $(\bar{\theta}, \{\mathbf{P}_i\}_{i=1}^N)$, we notice that the problem becomes tractable by fixing one set of the variables and solving for the other. Hence, we propose a two-step alternating procedure to solve Eq. (G.10): the *Merging* step and the *Aligning* step.

**Merging step.** In the merging step, we fix $\{\mathbf{P}_i\}_{i=1}^N$ and optimize for $\bar{\theta}$. This is equivalent to computing the mean of all individual transformed $\theta_i$ to estimate the merged model $\bar{\theta}$. Specifically, we solve

$$\left(\bar{\mathbf{A}}, \bar{\mathbf{B}}, \bar{\mathbf{C}}\right) \leftarrow \arg\min_{\bar{\theta}} \quad g\left(\bar{\theta}; \{\mathbf{P}_i\}\right), \text{ where}$$

$$\bar{\mathbf{A}} = \frac{1}{N}\sum_{i=1}^N \mathbf{A}_{\theta_i}^\top \mathbf{P}_i \mathbf{A}_{\theta_i}, \; \bar{\mathbf{B}} = \frac{1}{N}\sum_{i=1}^N \mathbf{P}_i^\top \mathbf{B}_{\theta_i}, \; \bar{\mathbf{C}} = \frac{1}{N}\sum_{i=1}^N \mathbf{C}_{\theta_i}\mathbf{P}_i.$$

Note that this is essentially the solution to some least-square problem.

**Aligning step.** In the aligning step, for each local dataset $i$, we fix the $\mathbf{A}_{\theta_i}$ and $\bar{\mathbf{A}}$, and compute the permutation matrix $\mathbf{P}_i$ by solving a linear assignment problem to align the local model with the merged model. To avoid the *two-sided* linear assignment problem that is not tractable, we reuse the permutation matrix from the *previous* iteration $\mathbf{P}_i'$ for the current iteration, so that we can solve it as a regular linear assignment problem (Bertsekas, 1998):

$$\min_{\mathbf{P}_i \in \mathcal{G}_{\text{perm}}} \quad \left\|\mathbf{A}_{\bar{\theta}} - (\mathbf{P}_i')^\top \mathbf{A}_{\theta_i}\mathbf{P}_i\right\|_F^2 + \left\|\mathbf{B}_{\bar{\theta}} - \mathbf{P}_i^\top \mathbf{B}_{\theta_i}\right\|_F^2 + \left\|\mathbf{C}_{\bar{\theta}} - \mathbf{C}_{\theta_i}\mathbf{P}_i\right\|_F^2 \quad (\text{G.11})$$

which can be simplified to solving

$$\max_{\mathbf{P}_i \in \mathcal{G}_{\text{perm}}} \quad \langle \mathbf{P}_i, \mathbf{A}_{\theta_i}^\top \mathbf{P}_i' \mathbf{A}_{\bar{\theta}} + \mathbf{B}_{\theta_i}\mathbf{B}_{\bar{\theta}}^\top + \mathbf{C}_{\theta_i}^\top \mathbf{C}_{\bar{\theta}}\rangle. \quad (\text{G.12})$$

We note that there are a finite number ($N^2$ pair) of pairs $(\mathbf{P}_i, \mathbf{P}_i')$, and the objective $\langle \mathbf{P}_i, \theta_1 P_{i-1}\theta_2^\top\rangle$ is non-decreasing. Therefore this alternating procedure is guaranteed to converge.

---

[6]Since there is only one hidden layer, for notational convenience, we use $\mathbf{P}_i$ as the permutation matrix for each local dataset $i$, instead of using $\mathcal{P}$ as in Section 2, which denotes a *sequence* of transformation matrices across layers. With a slight abuse of notation, we still use $\mathcal{G}_{\text{perm}}$ to denote the set of such permutation matrices and $\mathbf{P}_i \in \mathcal{G}_{\text{perm}}$. A similar convention applies to $\mathbf{P}_i \in \mathcal{G}_{\text{lin}}$ later to denote $\mathbf{P}_i$ belonging to the set of all invertible matrices.

### G.4.2 METHOD 2: UNCONSTRAINED GRADIENT DESCENT

Our second approach for policy merging is to find a solution in the set of general *invertible* matrices, through gradient descent for the objective as in Eq. (G.10) directly, i.e., now we allow $\mathbf{P}_i \in \mathcal{G}_{\text{lin}}$:

$$\min_{\bar{\theta}, \{\mathbf{P}_i\}_{i=1}^N \in \mathbb{GL}(n)^N} g\left(\bar{\theta}, \{\mathbf{P}_i\}_{i=1}^N\right) := \sum_{i=1}^N \left\|\mathbf{A}_{\bar{\theta}} - \mathbf{P}_i^{-1}\mathbf{A}_{\theta_i}\mathbf{P_i}\right\|_F^2 + \left\|\mathbf{C}_{\bar{\theta}} - \mathbf{P}_i^{-1}\mathbf{B}_{\theta_i}\right\|_F^2 + \left\|\mathbf{C}_{\bar{\theta}} - \mathbf{A}_{\theta_i}\mathbf{P_i}\right\|_F^2 .$$
(G.13)

We consider a simpler loss, which is obtained by right-multiplying the expression inside the first and second terms by $\mathbf{P}_i$:

$$\min_{\bar{\theta}, \{\mathbf{P}_i\}_{i=1}^N \in \mathbb{GL}(n)^N} g\left(\bar{\theta}, \{\mathbf{P}_i\}_{i=1}^N\right) := \sum_{i=1}^N \|\mathbf{P}_i\mathbf{A}_{\bar{\theta}} - \mathbf{A}_{\theta_i}\mathbf{P_i}\|_F^2 + \|\mathbf{P}_i\mathbf{B}_{\bar{\theta}} - \mathbf{B}_{\theta_i}\|_F^2 + \|\mathbf{C}_{\bar{\theta}} - \mathbf{C}_{\theta_i}\mathbf{P_i}\|_F^2 .$$
(G.14)

Note that the losses (G.13) and (G.14) are equivalent in spirit in the sense that, if there exists matrices $\{\mathbf{P}_i\}$ which perfectly align all $\theta_i$, then those same matrices are the global optima for both expressions. The second expression (G.14) has the advantage of admitting simpler gradients, and is individually convex with respect to either $\{\mathbf{P}_i\}$ or $(\mathbf{A}_{\bar{\theta}}, \mathbf{B}_{\bar{\theta}}, \mathbf{C}_{\bar{\theta}})$, holding the other fixed. Therefore, for $N$ models, we run alternating gradient descent to update those parameters to update the averaged model parameters $\bar{\theta}$ and the transformation matrices $\mathbf{P} = (\mathbf{P}_1, \cdots, \mathbf{P}_N)$.

**Remark G.1** (Possibility of Degenerate Solutions). Note that Eq. (G.14), as long as $\mathbf{B}_{\theta_i}$ are non-zero, the degenerate solution with $\mathbf{P}_i = 0$ will not be optimal even in Eq. (G.14). More generally, we find that gradient descent on Eq. (G.14) works well in practice. An alternate approach would be to alternate gradient updates on $\{\mathbf{P}_i\}$ with exactly solving for the optimal $\bar{\theta} = (\mathbf{A}_{\bar{\theta}}, \mathbf{B}_{\bar{\theta}}, \mathbf{C}_{\bar{\theta}})$ in Eq. (G.14) (which is a well-conditioned least squares problem whenever the singular values of each $\mathbf{P}_i$ are bounded away from zero). In our experiments, we find this is not necessary.

### G.5 EXPERIMENT SETUP DETAILS

We set up a linear system in the experiment with state dimension $n = 4$, input dimension $m = 2$, and

To model different tasks, we choose $\mathbf{R} = \mathbf{I}_2, \mathbf{Q} = \alpha^{(h)}\mathbf{I}_4$ with $\alpha^{(h)} \in \log \text{space}(-2, 2, H + 1)$ and $H = 9$.

We consider such 10 different system realizations, and depending on whether it is partially observable (LQG case) or fully observable (LQR case), we choose the observation matrix $\mathbf{C}$ as a randomized matrix (but fixed for each task) by sample each entries from a unit gaussian or $\mathbf{C} = \mathbf{I}$, respectively. For the former, the entries in $\mathbf{C} \in \mathbb{R}^{50 \times 4}$ are independently sampled from a Gaussian distribution. We run 10 trials for each experiment. The rollout horizon is 100. The system dynamics follow from that in Hong et al. (2021), and the setup of $\mathbf{R}, \mathbf{Q}$ follows from that in Zhang et al. (2022) on multi-task imitation learning in linear control. We generate expert data with the optimal LQG/LQR controller, and train the policies with imitation learning. Policy performance is evaluated in terms of the closed-loop rollout cost. Finally, in both the settings of learning static and dynamic policies, the parameterized controllers are trained with a standard gradient descent algorithm.

For the last experiment, we set up the following systems with 10 systems where half of them are positive and half of them are negative, following the disconnected component constructions in the next section. We show that under disconnected component settings, we need general invertible matrices to successfully merge different linear policies.

$$\mathbf{A}_{\text{unstab}} = \begin{bmatrix} 1.1 & 0.03 & -0.02 \\ 0.01 & 0.47 & 4.7 \\ 0.02 & -0.06 & 0.40 \end{bmatrix}, \quad \mathbf{B} = \begin{bmatrix} 0.01 & 0.99 \\ -3.44 & 1.66 \\ -0.83 & 0.44 \\ -0.47 & 0.25 \end{bmatrix}.$$

### G.6 RESULTS AND DISCUSSIONS

### G.6.1 BASIC EXPERIMENTAL RESULTS

As shown in Figure 13, with *static* feedback policies (i.e., "feedforward" policies), *average* can sometimes match the performance of policies that are trained with shared data (*multi-task*) or task-

specific data (*task-specific*). We then compare the performance in cases with *dynamic* policies (i.e., "recurrent" policies), and observe that the gradient-based method can outperform the alternation-based method that searches in only the space of permutation matrices. These two methods, by accounting for the similarity invariance in merging, outperform the naive averaging method (*average*). They also outperform *single-task*, where policies are only trained with local datasets, since more data has been (implicitly) used by merging multiple policies. More interestingly, when there are disconnected components in the landscape (as constructed in odd-dimensional systems, see more details in the next subsections), we observe that only searching for permutation will not mitigate the issues.

### G.6.2 GRADIENT-BASED APPROACH V.S. LQG LANDSCAPE

In this subsection, we reconcile the effectiveness of gradient-based policy alignment with the topological properties of the the LQG cost landscape. First, recall that we parameterize controllers by $\pi = (\mathbf{A}_\theta, \mathbf{B}_\theta, \mathbf{C}_\theta)$, where $\mathbf{A}_\theta$ maintains a latent state $\hat{\mathbf{x}}$ of the same dimension as the system state. Let $\mathcal{C}_n$ denote the set of policies $\pi = (\mathbf{A}_\theta, \mathbf{B}_\theta, \mathbf{C}_\theta)$ such that the cost $J(\pi)$ is finite, or equivalently, the set of controllers which *stabilize* the dynamics (Zhou et al., 1996). Zheng et al. (2021) show that this set is either (a) fully connected, or (b) disconnected, but has only two connected components $\mathcal{C}_n^{(1)}, \mathcal{C}_n^{(2)}$. Case (a) occurs if either

  (i) the *open loop system* is stable, i.e. the spectral radius $\rho(\mathbf{A}) < 1$, that is i.e. maximum eigenvalue magnitude) of the system dynamic matrix is strictly less than 1. This is equivalent to saying that $J_T(\mathbf{0})$ is finite for the zero controller $(\mathbf{A}_\theta, \mathbf{B}_\theta, \mathbf{C}_\theta) = (\mathbf{0}, \mathbf{0}, \mathbf{0})$.

  (ii) there exists a policy $\pi$ with rank$(\mathbf{A}_\theta) < n$ such that $\pi \in \mathcal{C}_n$.

And, in case (b), Zheng et al. (2021, Theorem 3.2) shows that any two connected components can be realized by any mapping of the form

$$(\mathbf{A}_\theta, \mathbf{B}_\theta, \mathbf{C}_\theta) \mapsto (\mathbf{T}\mathbf{A}_\theta\mathbf{T}^{-1}, \mathbf{T}\mathbf{B}_\theta, \mathbf{C}_\theta\mathbf{T}^{-1}) : \det(\mathbf{T}) < 0. \tag{G.15}$$

For example, in odd dimensions, the matrix $\mathbf{T} = -\mathbf{I}$ has $\det(\mathbf{T}) = -1$. Thus, we find a simple corollary:

**Proposition G.1.** Let the state dimension $n$ be odd. Then, if $\mathcal{C}_n$ is disconnected and $\theta = (\mathbf{A}_\theta, \mathbf{B}_\theta, \mathbf{C}_\theta)$ is stabilizing, then the policies $(\mathbf{A}_\theta, \mathbf{B}_\theta, \mathbf{C}_\theta)$ and $(\mathbf{A}_\theta, -\mathbf{B}_\theta, -\mathbf{C}_\theta)$ lie in different connected components of $\mathcal{C}_n$. More generally, consider any block diagonalization of a

$$\begin{bmatrix} \mathbf{A}_\theta^{(1,1)} & \mathbf{A}_\theta^{(1,2)} \\ \mathbf{A}_\theta^{(2,1)} & \mathbf{A}_\theta^{(2,2)} \end{bmatrix}, \begin{bmatrix} \mathbf{B}_\theta^{(1,1)} \\ \mathbf{B}_\theta^{(2,1)} \end{bmatrix}, \begin{bmatrix} \mathbf{C}_\theta^{(1,1)} & \mathbf{C}_\theta^{(1,2)} \end{bmatrix}, \tag{G.16}$$

where the block dimensions are chose such that "1" dimension has odd dimension. Then, if the above is stabilizing and $\mathcal{C}_n$ is disconnected, then the following equivalent controller lies in the other connected component of $\mathcal{C}_n$:

$$\begin{bmatrix} \mathbf{A}_\theta^{(1,1)} & -\mathbf{A}_\theta^{(1,2)} \\ -\mathbf{A}_\theta^{(2,1)} & \mathbf{A}_\theta^{(2,2)} \end{bmatrix}, \begin{bmatrix} -\mathbf{B}_\theta^{(1,1)} \\ \mathbf{B}_\theta^{(2,1)} \end{bmatrix}, \begin{bmatrix} -\mathbf{C}_\theta^{(1,1)} & \mathbf{C}_\theta^{(1,2)} \end{bmatrix}. \tag{G.17}$$

As a second example, note that when $\mathbf{T}$ is an "odd" permutation, $\det(\mathbf{T}) = -1$, placing the transformed parameters into separate components.

### G.6.3 INSUFFICIENCY OF NAIVE AND PERMUTATION BASED MERGING

Importantly, *path connectedness* does not mean connectedness by *linear paths*, the latter being precisely the definition of convexity. Indeed, consider the equivalence transformation $(\mathbf{A}_\theta, \mathbf{B}_\theta, \mathbf{C}_\theta) \mapsto (\mathbf{A}_\theta, -\mathbf{B}_\theta, -\mathbf{C}_\theta)$. The mid point between these controllers if $(\mathbf{A}_\theta, \mathbf{0}, \mathbf{0})$ which, one can check, is equivalent to the zero controller $(\mathbf{0}, \mathbf{0}, \mathbf{0})$. Not only is the zero controller essentially *never* optimal, if $\mathbf{A}$ is not stable, then this controller is not stabilizing. Similarly, permutations can be insufficient to merge as well: indeed, permutation based mergin coincides with naive merging in dimension $n = 1$ one, which the above shows will fail if $\mathbf{A}$ is unstable (i.e., in dimension one, $|\mathbf{A}| \geq 1$). These examples highlight the important of aligning using *all* invertible matrices, and not simply permutation matrices.

### G.6.4 AVOIDING PATH-DISCONNECTNESS OF STABILIZING CONTROLLERS

In addition to the insufficiency of linear and permutation-based merging, it may seem that the possible *path disconnectedness* of the set $\mathcal{C}_n$ may pose a challenge to weight merging. However, are experiments do not seem to suggest that this is an issue for the unconstrained gradient descent method proposed in Appendix G.4.2. The reason for this is our choice of merging cost in (G.14). This cost penalizes only differences between the *controller parameters*. It does not, as in FLEET-MERGE (Algorithm 1), use a cost depending on the policy error. Therefore, even if one of the iterates lies outside of the stabilizing set $\mathcal{C}_n$ (and therefore has *infinitely large* infinite-horizon LQR cost $J$), we can still compute gradients with respect to the *parameter error* (G.14). In other words, merging in parameter loss allows us to pass through disconnectedness in the landscape of control cost $J$.

This can be seen in the following simplified variant of (G.14) for the merging of two LQG policies $\theta_1 = (\mathbf{A}_1, \mathbf{B}_1, \mathbf{C}_1)$ and $\theta_2 = (\mathbf{A}_2, \mathbf{B}_2, \mathbf{C}_2)$. Define

$$\mathcal{L}(\mathbf{P}) = \|\mathbf{P}\mathbf{A}_1 - \mathbf{A}_2\mathbf{P}\|_{\mathrm{F}}^2 + \|\mathbf{B}_1\mathbf{P} - \mathbf{B}_2\|_{\mathrm{F}}^2 + \|\mathbf{C}_1 - \mathbf{C}_2\mathbf{P}\|_{\mathrm{F}}^2. \tag{G.18}$$

Then note that this loss is convex in $\mathbf{P}$. Under natural conditions, it is in fact *strongly convex*, and thus admits a unique global minimizer $\mathbf{P}$. If $\theta_1$ and $\theta_2$ are equivalent, i.e., $(\mathbf{A}_2, \mathbf{B}_2, \mathbf{C}_2) = (\mathbf{T}\mathbf{A}_1\mathbf{T}^{-1}, \mathbf{T}\mathbf{B}_1, \mathbf{T}\mathbf{C}_1)$ for some transformation $\mathbf{T}$, then this unique minimizer must be $\mathbf{P} = \mathbf{T}$, as this has zero loss in (G.18). This is true regardless of whether or not $\theta_1, \theta_2$ lie in separate connected components of $\mathcal{C}_n$. Hence, we see how merging based on parameters alone circumvents possible loss disconnectedness.

