# OpenReview forum: "Robot Fleet Learning via Policy Merging"
_ICLR.cc/2024/Conference — ICLR 2024 poster_

### Official Review · Reviewer_DxkE · 2023-10-30

**Soundness:** 4 excellent
**Presentation:** 4 excellent
**Contribution:** 2 fair
**Rating:** 8
**Confidence:** 3

**Summary:**

The authors propose a method to merge policies learnt by a fleet of robots trained for different skills. The proposed FLEET-MERGE algorithm which allows for distributed learning of recurrent neural networks through a merging strategy that accounts for permutation invariance of neurons in layers. The authors also introduce a new robot manipulation benchmark for fleet policy learning, which they use to evaluate their method.

**Strengths:**

1) The experimental results for policy merging are strong for merging RNN policies trained for different tasks. The proposed method outperforms naive averaging and other similar baselines. Distributed learning is becoming increasingly important for robotics as deployment of fleets of robots for data collection is more feasible.

2) The introduced benchmark FLEET-TOOLS is a useful contribution to the robot manipulation community, for easy collection of expert trajectories useful for training policies with imitation learning.

3) The algorithms and experiments are explained clearly, with good presentation.

**Weaknesses:**

1) The proposed algorithm was only demonstrated for behavior cloning, which is not what we generally consider as the setup where we benefit from distributed learning. If we have a large amount of static expert data, it is not too difficult to just merge the datasets and train a policy on the joint dataset. When we consider fleet policy learning, it is much more useful to consider a reinforcement learning setup where a collection of robots are collecting data for various tasks, and we wish to collectively use these data sources to learn optimal policies. But this setup is much more complicated due to non-stationarity and other challenges with online RL. It would be much more interesting to see if these merging strategies could work in an online RL setting.

2) The novelty of the proposed method is not very clear. The algorithm used by the authors is based heavily on Pena et al. [1], as such it is hard to see what the main contribution is, apart from the FLEET-TOOLS benchmark. Is it the application of the algorithm for training RNNs, and the experimental demonstration of its usefulness in fleet robot learning?

**Questions:**

1) Could the authors motivate distributed policy learning for behavior cloning? If static expert datasets are already present, it does not seem useful to train separate policies and merge them. Does this approach work in the online reinforcement learning setting?

2) Could the authors summarize the main novel contributions of the FLEET-MERGE algorithm, especially compared to the work from Pena et al. [1]? I would be content if the contributions are mainly experimental validation for robot learning, but would appreciate the clarity since the exact contribution is difficult for me to parse.

It is possible I did not understand the contributions properly, so I would be willing to raise the score if the authors can provide a strong response to my questions. But as of now I am leaning slightly towards rejection since behavioral cloning does not seem to be an appropriate task to motivate fleet learning, and the novelty of the FLEET-MERGE algorithm is not clear.

[1] Re-basin via implicit Sinkhorn differentiation, arxiv.org/abs/2212.12042

---

> ### Author Response · Authors · 2023-11-15
> **Thank you for your constructive feedback!**
>
> Thank you for finding the topic of our work is of “useful contribution to the robot manipulation community” and “increasingly important” , and appreciating our contribution. We address your questions below.
>
>
> **Behavior Cloning:**
>
> Thank you for this comment. Although there are a plethora of policy-training techniques, we focus on behavior cloning (BC) for the following main reasons:
>
> > 1) Its simplicity helps us disambiguate the **effects of our method** from the **intricacies** of the underlying training methodology’’.
>
> >  2) BC is arguably the state-of-the-art approach for the **industry-scale robot policy training** settings where our method is most useful [1].
>
> >  3) Even with static/offline datasets, as in BC, data-pooling is *not* always feasible. In particular, the datasets for **policy learning** in robotics are usually videos and embodied robotic data (such as observation-action trajectories), compared to those for **supervised learning** (as texts), can be much heavier to communicate and centralize. Moreover, robots belonging to different service providers/companies/client industries/households are not motivated/have the liberty to share the local data that can contain **private** information of the clients (analogous to the privacy concerns of federated learning for mobile devices).
>
> [1] Open X-Embodiment Collaboration. Open X-Embodiment: Robotic learning datasets and RT-Xmodels. https://robotics-transformer-x.github.io, 2023
>
>
> We do agree that it is interesting to see if our method extends to online RL approaches, and we acknowledge the reviewer’s point that the benefits of distributed learning are *even more pronounced* for online RL training. As per your suggestion, we have included some preliminary experiments in the end of the revised paper (Sec. E in Appendix), demonstrating that yes, indeed, model-merging approaches in general are effective in the RL setting as well and are robust even with little communications.
>
>
> **Algorithmic Novelty**:
>
> We want to emphasize that the proposed Fleet-Merge algorithm is different from a simple extension of Pena et al., in the following important aspects:
>
> >  1) We are dealing with the “policy merging” setting with *multiple models* rather than just *aligning two models*, as considered in Pena et al.;
>
> > 2) Because of 1), our algorithm maintains a central model as the “anchor policy” in merging – as in the seminal FedAvg algorithm – which allows for an “interpolation” between “pure model aligning (with one-shot merging)” and “federated learning (with multiple rounds of merging)”. We consider this as one of our key algorithmic and conceptual contributions. In contrast, a naive application of Pena et al. would require **pairwise** aligning and merging, as in the GitRebasin paper, which may depend on the order of the merging sequence;
>
> >  3) We include a “hard projection” step in our algorithm, which takes on a new interpretation. Rather than permuting the weights to evaluate mode connectivity, we use hard projections as a way to compute optimal mode alignment for merging. We show via our ablations (Figure 12 in the appendix) that this is necessary;
>
> >  4) Our algorithm allows *iterative merging*, i.e., merging multiple times during the course of training, compared to other (feedforward) neural network aligning approaches as in Pena et al., which allows *tradeoffs between communication cost and merged-model performance*;
>
> >  5) Unlike the two-model-merging setting of Pena et al., we do not have access to a common dataset $\mathcal{D}$ for the aligning updates. Instead, each model is updated by sampling trajectories from its *local dataset* $\mathcal{D}_{local,i}$, obviating the need for dataset sharing (the key consideration in our setup).
>
> >  6) Finally,  because we consider  *policy learning* from *partial observations* as images/point-clouds, the models are parameterized by *recurrent* neural networks, forcing us to handle permutation symmetries not only between *layers*, but also between *timesteps* which was not dealt with in Pena et al., which only handled feedforward neural networks;
>
>
> Thank you very much again for your feedback, and please do let us know if you have any other questions and/or comments.

---

> > ### Author Response · Authors · 2023-11-20
> > **Thank you for your constructive feedback!**
> >
> > Dear Reviewer, we would like to thank you again for your efforts and time in providing thoughtful feedback and comments. We’ve revised the paper according to your suggestions and replied to all the questions and concerns. Since the discussion period is ending soon, we would greatly appreciate it if you could let us know whether you have any additional comments. Thanks a lot!

---

> > ### Comment · Reviewer_DxkE · 2023-11-21
> >
> > I thank you for your detailed response, and the additional RL experiments. I think the simple cartpole RL experiment is insufficient to show if fleet-merge will work in general RL training for more complex tasks. However, the addition is still appreciated. I am also satisfied with the explanation for the novelty of Fleet-Merge.
> >
> > I still have not been convinced that fleet-learning can be useful in the behavior cloning context, however I do not think this is a significant issue with the paper. As such I am raising my score to 8. I encourage the authors to add more RL experiments in more complex tasks.

---

> > > ### Author Response · Authors · 2023-11-21
> > > **Thank you again for your constructive feedback!**
> > >
> > > Dear Reviewer, Thank you very much for the feedback!

---

### Official Review · Reviewer_2Z2Z · 2023-10-31

**Soundness:** 3 good
**Presentation:** 2 fair
**Contribution:** 3 good
**Rating:** 5
**Confidence:** 4

**Summary:**

This paper studies policy merge, which seeks to merge policies that are trained on different datasets and tasks into a single set of weights, while preserving the learned skills of the original policies. In particular, taking account into the sequential and dynamic nature of robot policy learning, this paper focuses on merging RNN-parameterized policies. The proposed method, Fleet-Merge, outperforms prior model merging methods by over 50%, accounting for not only the permutation symmetries in RNN policies but also enabling multiple rounds of merging between each training update as well as merging multiple models.

**Strengths:**

This paper is very novel in both its problem setting. Policy merging, to the best of my knowledge, has not been explicitly studied before. This paper will have impact in the robot learning community, where large-scale datasets as well as data/model sharing across institutions are becoming increasingly prevalent. However, most such effort has primarily focused on consolidating all data to train a single large model (e.g., the recent RT-X effort), but this paper offers a fresh perspective that opens up the possibility of different institutions sharing their pre-trained policies, which are much more manageable.

The proposed method, at a technical level, is largely derived from a prior work (Pena et al., 2022). That said, the paper does introduce several additional improvements that are particular fitting to the policy learning setting. For one, Fleet-Merge is designed to be able to merge multiple models and do so in multiple-round fashion, both of which are important features to have in sequential decision making fashion; in contrast, prior works in the model merging literature is primarily considering the simplified case of merging two models. Because of these changes, Fleet-Merge does diverge from prior works with several modifications such as grounding the model merging in each iteration to the current average of all models, reminiscent of common practices in federated learning. Therefore, the approach, though extensions of prior works, is well motivated.

**Weaknesses:**

The main weaknesses of the paper is in its presentation and the strength of the experimental results.

First, the presentation of the paper can be improved. In Figure 3 and 4, several charts are hard to see because the legends block them.  The experiment section can be better structured by having subsections for each of the evaluation setting instead of each benchmark; right now, all evaluation settings are presented upfront without concrete contextualization and results that interleave between them. The new benchmark Fleet-Tools is not well motivated and the section on "Keypoint transformation & trajectory optimization" within Section 4 is too dense while not directly related to the algorithmic components of the paper. I suggest the authors better motivate why Fleet-Tools is particularly well-suited for studying the policy merging problem.

The performance improvement of Fleet-Merge compared to simple baselines is moderate at best. Fleet-Merge does not seem to be substantially better than Git-Rebasin or FedAvg on the Metaworld and Fleet-Tool benchmark.  Considering that either baseline is missing key components in Fleet-Merge, it is worth asking the question whether Fleet-Merge is truly necessary given its added algorithmic complexity as well as computational cost. Furthermore, while the paper emphasizes the multi-task aspect of policy merging, most of the experiments are conducted in the single-task setting. Interestingly, in the multi-task setting, simple averaging appears to be very competitive.

Overall, I think this paper is promising and I am willing to improve my score if the authors could resolve my reservations about the practical utility of the proposed method as well as more clearly present the paper.

**Questions:**

Several suggestions and questions are already included in the Weaknesses section. Here are some additional questions/suggestions:

1. The definition of Loss Barrier should be included somewhere early. In Section 5, the concept of "performance barrier" is introduced assuming readers' familiarity with the equivalent concept of loss barrier in the supervised learning model merging literature.

---

> ### Author Response · Authors · 2023-11-15
> **Thank you for your constructive feedback!**
>
> Thank you for finding our work **very novel**, and appreciate the potential **impact in the robot learning community**. We also appreciate your comments on presentation and experiments, and have addressed them in the revision and the responses below.
>
> **Presentation**. We appreciate all your constructive feedback regarding our presentation, and have incorporated them in the revision. We summarize our changes as follows:
>
> >  (a) We have updated the charts in Figures 3 and 4 as suggested, making them clearer.
>
> >  (b) We have revised the Experiments section by restructuring it according to Experimental Settings instead of Experimental Benchmark, as per the reviewer’s suggestion.
>
> >  (c) Thanks for bringing up the confusion about the definition of **loss barrier**. We have added the definition earlier in Sec. 3, as the reviewer suggested.
>
>
> >  (d) We believed the design of Fleet-Tools, as a novel benchmark for robotic manipulation tasks that allows easy construction of compositional and contact-rich tasks, is an important contribution to robot (fleet) learning in its own right (in addition to our **algorithmic components**).  This is why we endeavored to provide a detailed introduction to the benchmark, though we recognize that the writing was somewhat dense, so we have revised it to be clearer.  We believe that this benchmark is well suited to testing fleet policy learning for the following reasons:
>
> >> (i) the ability to adjust proportions of combinations  of objects and tools provides a systematic and scalable way to create **local/distributed datasets**  in fleet learning
>
> >> (ii) the Drake simulator allows for physically realistic dynamics, giving us a better sense of the efficacy of merging in the real world.
>
> >> (iii) Our benchmark allows for rapid construction of  **local demonstration data** that readily enables high-fidelity policy learning.
>
> >> (iv) The tool-use tasks considered in Fleet-Tools are more challenging than common tasks in most existing robotic manipulation simulators, e.g., MetaWorld (for which we have tested our framework also). To the best of our knowledge, Fleet-Tools is the first benchmark designed for imitation learning that involves tasks with intricate and challenging dynamics.
>
> As per the reviewer’s feedback, we have simplified the introduction of Fleet-Tools, and added more discussions on its motivation for fleet learning, in the main body of the paper.
>
>
> **Fleet-Merge performance**. Thank you for this valuable feedback. We have contextualized the relative increases in performance in the revision. In short, robotic behavior cloning tasks are quite challenging and obtaining *any* increase in performance should be considered a significant achievement. Notice, for example, that an increase from a score of .55 to .65, as observed in Figure 3(d) with alpha = .01 corresponds to a *20% gain* in performance, which we consider to be significant. Moreover, while there is indeed some additional algorithmic complexity introduced by Fleet-Merge, the additional computational overhead is minor. This is because merging operations are *infrequent* in our overall setup, and thus the cost of merging is dominated by the training time. Beyond the performance improvements, we believe that the conceptual contribution of this paper -- *that of model merging over data sharing* — will inspire future algorithms to achieve even greater performance improvements as others in the community consider this important problem of fleet policy learning.
>
> **Multi-Task**. We acknowledge that on the Meta-World benchmark, simple averaging is surprisingly competitive. Notice that this too is an interesting finding, and we view it as a contribution of our empirical study – *merging **models** instead of **data** can work in multi-task robot learning*, rather than a drawback of our proposed approach. However, we believe that the Meta-World experiments exactly motivate our Fleet-Tools benchmark for fleet learning: whereas simple merging may suffice for the former, Fleet-Merge is necessary to handle the additional dexterity for robotic manipulation demanded by the latter. We further note that Fleet-Tools can be viewed as another form of multi-task learning, wherein the combination of each object to be manipulated and the tool used corresponds to a **task**. We have made such a connection clearer in the revision. We believe that task- or language-conditioning policies will be interesting future work.
>
> Thank you very much again for your constructive feedback. In sum, we have revised the paper significantly by adding more motivations of our setup and benchmark, as well as restructuring the sections and introducing additional explanations to improve the paper presentation. Please do not hesitate to let us know if you have any other questions and/or comments.

---

> > ### Author Response · Authors · 2023-11-20
> > **Thank you for your constructive feedback!**
> >
> > Dear Reviewer, we would like to thank you again for your efforts and time in providing thoughtful feedback and comments. We’ve revised the paper according to your suggestions and replied to all the questions and concerns. Since the discussion period is ending soon, we would greatly appreciate it if you could let us know whether you have any additional comments. Thanks a lot!

---

> > > ### Comment · Reviewer_2Z2Z · 2023-11-22
> > >
> > > Dear Authors,
> > >
> > > Thank you for your detailed responses! I have a few thoughts:
> > >
> > > 1. Figure 3(d) doesn't seem to suggest an improvement from 0.55 to 0.65. Could you clarify?
> > >
> > > 2. I am still not sure about the value of the new benchmark. While Fleet-Tools is presented as a challenging benchmark that involves rich challenging tool uses and contact dynamics, all methods presented in the paper appear to have higher relative success rate on Fleet-Tools tasks compared to the ones in MetaWorld. The difference between Fleet-Merge and baselines also do not appear to be large (no error bars presented in Figure 4 as well). Given these, while I commend that newer benchmarks are introduced, I still do not believe that this benchmark is sufficiently motivated or useful in the context of this paper.
> > >
> > > 3. The set of baselines in MetaWorld and Fleet-Tools are not consistent. Across Figure 3 and 4, some methods/experiments are run only on one benchmark versus other. It's not clear the justification behind this choice.

---

> > ### Author Response · Authors · 2023-11-22
> > **Any other suggestions?**
> >
> > Dear Reviewer 2Z2Z,
> >
> > Hope all is well! Since today is the last day of the author-reviewer discussion phase (and we have not heard back from you), we were wondering if our revision of the paper and response have addressed your previous comments. Please do not hesitate to let us know if there are any other suggestions/comments.
> >
> > Thank you very much!
> >
> > The authors

---

> ### Author Response · Authors · 2023-11-22
> **Thank you for your constructive feedback!**
>
> Thanks again for your feedback, and we will update the draft to reflect these changes.
>
> > Figure 3(d) doesn't seem to suggest an improvement from 0.55 to 0.65. Could you clarify?
>
> Sorry for the typos in the response due to the reorganization of the paper (and subfigures). The improvement should refer to Figure 4 (a,b more specifically) where Fleet-Merge outperforms other methods (and apologize that the exact number should be from 0.44 to 0.65, which is even better). We also highlight that in the supervised learning setting in Appendix F Figure 12, Fleet-Merge can have over 100% improvements compared to FedAvg in some cases.
>
> >  I am still not sure ... in the context of this paper.
>
> While we agree MetaWorld is a useful benchmark in its own right, we argue that it is necessary to test on a *second suite of tasks* to confirm that the benefits of Fleet-Merge do not overfit to the specificities of any single environment. We took care to make our methodology for the construction of Fleet-Tools transparent so that it is clear that we *did not* in any sense tune or overfit this environment to improve the relative performance of Fleet-Merge. Rather, it is because Fleet-Tools tasks are *intrinsically more challenging*  (as described below, and as in our previous response) than those on Meta-World that we have room to see greater relative gains in performance for Fleet-Merge over comparator methods. Also, as mentioned before, the combinatorial composition of tasks in Fleet-Tools makes it scalable and easy to create local datasets in fleet learning.
>
> We acknowledge that Fleet-Tools is not limited to only study fleet learning, but is more general for robotic manipulation. Nevertheless, we believe this benchmark provides a number of advantages:
>
> (1) Tool-use is a challenging task that requires high dexterity and generalization. This allows us to test methods on more challenging tasks.
>
> (2) While there may ultimately be abundant data for these tasks in real robots and human demonstrations in tool-use, we still lack simulation data for effective benchmarking of robotic tool-use tasks. Fleet-Tools provides a purely in-simulation benchmark.
>
> (3)  Tool-use is particularly well-suited for studying multitask policies or policy learning under heterogeneity, because it requires object-object affordance, and tool-object compositionality (humans can use one tool for many tasks). This multitask flavor is useful for studying the ability of Fleet-Merge to combine diverse policies, and becomes especially relevant in fleet-learning scenarios.
>
> (4) Fleet-Tools is the first robotic benchmark built with *Drake*. Drake offers numerous unique benefits as a simulation environment that complements other simulators such as Mujoco or IsaacGym: these features include trajectory optimization, model-based control, optimization, and the code is open source. Our Fleet-Tools thus inherits these advantages as a simulation benchmark.
>
>
> > The set of baselines ... It's not clear the justification behind this choice.
>
> All benchmarks in the paper serve slightly different purposes. We made sure to run all experiments which “made sense” on either benchmark. Specifically, we study *linear mode connectivity* for both Meta-World and Fleet-Tools in Figure 3 (a,b), and *robustness to communication frequencies* in Figure 4 (a,d). However, there were some experiments that were better suited to one benchmark versus the other. The choice of results reported in Figures 3 and 4 is purely a matter of presentation. We will explain these decisions here and then be sure to clarify further in the revision:
>
> (1) Meta-World is a relatively simple standard benchmark and is used as a proof-of-concept. It can run faster and in larger-scale experiments; it also has many tasks to choose from, which means policies training on these separate tasks can be very diverse. Therefore, it is well-suited to testing the effects of changing the fraction of policies selected for merging in Fleet-Merge at each round (Algorithm LIne 6). This is what has been reported in Figure 4c.
>
> (2) Fleet-Tools is more challenging and closer to the real-world setup. Thus, we were able to include real-world figures and some preliminary sim-to-real results in Appendix Figure 7. Because of this, we used Fleet-Tools to evaluate experiments that would be more indicative of real-world setups. This motivated our use of Fleet-Tools in the fine-tuning experiments (Figure 3c). We will be sure to include fine-tuning results on Meta-World as well for the camera-ready.
>
> As described in the paper, we investigate other ablations with our Linear control benchmark and supervised learning experiments.
>
> Thank you very much again for the feedback! Hope our responses and revisions have addressed your remaining concerns, especially regarding "practical utility of the method" and "more clearly present the paper" as you have commented before. Feel free to let us know if there is anything else that can help improve the paper.

---

### Official Review · Reviewer_FoVN · 2023-11-01

**Soundness:** 4 excellent
**Presentation:** 4 excellent
**Contribution:** 4 excellent
**Rating:** 8
**Confidence:** 3

**Summary:**

This paper introduces a method, called Fleet-Merge, for merging neural network policies trained on different tasks into one neural network policy. When a large fleet of robots learn a diverse set of skills on diverse tasks and it is not possible to collect a centralized dataset of all environmental interactions, how can we learn singular policies that capture all the training done by individual robots? A common tool to merge policies in federated learning is to simply average the weights/biases of all the neural networks into one policy. However, this leads to a drop in performance because the internal representation space of these policies may be significantly different. This paper introduces a method for merging the policies using the permutation invariance property of RNNs. By applying a permutation matrix to each layer of weights and biases, the paper aligns the internal representation space of the RNNs and is able to perform better merging. The authors test on a series of domains including linear control tasks and realistic robotic control tasks. They also introduce a tool use benchmark called Fleet-Tool, a task suite for robotic tool usage (like hammering, screwing, cutting,...)

**Strengths:**

This is a very interesting paper and is well written. The problem that they investigate is well supported and they propose (to my knowledge) a novel algorithm that allows us to merge different neural networks by aligning their internal representation spaces. They do extensive experiments in simulation and the real-world and show that policies from diverse tasks can be merged together in a smart way.

They leverage known properties of RNNs (i.e.  internal permutation invariance and time-invariance) and find transformation matrices to match the parameters of different RNNs together. This is an interesting idea and very applicable to the problem that they are solving.

**Weaknesses:**

While the method proposed is clear to me, I do not understand how the merged policy can determine what task to solve during test time.

Further, in the results section I did not see how well the policies perform on a single task. In figure 4, it is clear that FleetMerge performs better than Federated Averaging, however I am curious to see how much of a decrease in performance is incurred by merging the policies at all.

**Questions:**

1. At test time, how does the merged policy know what task to solve? Could there be an ambiguity on what the task specification is?
2. What would the performance be of a policy trained on all the data collected during the fleet learning? I am curious to know what the upper bound on multi-task training is.
3. When choosing the first permutation matrix, does it matter which RNN  policy you align the rest of the policies? I am unclear on how this initialized permutation matrix is chosen/calcualted

---

> ### Author Response · Authors · 2023-11-15
> **Thank you for your constructive feedback!**
>
> Thank you for finding the topic of our work very interesting and novel, and the paper well-written. We address your questions as follows.
>
>
> > **Which task to solve:** This is a good point and we have added clarification in the revision. We consider multi-task settings in which the task can be inferred from observations. For example, in Fleet-Tools, the task is always to apply a tool to the given object, and the tool and object can be inferred from the point cloud, which is the input of the policy.  For both Fleet-Tools and Metaworld, we verify that there exists a single policy which executes all the tasks with proficient performance. We have included plots of this centralized trained policy in the Appendix. We also note that our method is entirely consistent with task- or language-conditioned policies, and this remains an exciting direction for future work. If we do not know what the test-task is, we can envision Fleet-Merge as a tool for training a large-pretrained model, which can be refined further via *finetuning*. Indeed, our experiments (Figure 3(b)) show that Fleet-Merge is effective for this purpose.
>
> > **Performance of training on all data:** Thanks for bringing this up. We have added the **upper-bound** performance of the joint policy training to figures and mentioned in captions as “the performance is upper-bounded and lower-bounded by the joint policy success rates and one-shot policy merging.” We observe that there is some degradation of the performance depending on the frequencies of (thus the number of times of) averaging, but overall the weight merging works surprisingly well. More specifically, in Figure 3 describing the wrench task in Fleet-Tools, the upper bound is around 0.82 success rate if we pool all the task data together. For Metaworld, the joint training success rates often can be really high (around 0.9). We also like to emphasize that *we do have multitask results* and one can also check the Metaworld experiments in the Appendix for larger-scale experiments.
>
> > **Choice of initial permutation:** The permutation matrices are initialized as identity and the initial anchor policy is the “naive averaged” policy. We attempted to improve performance by experimenting with more advanced initialization schemes, but none improved performance to a degree warranting the additional complexity. We describe this detail in *Algorithm description* in Sec. 3.1.
>
> Thank you very much again for the positive comments, and hope our responses have fully addressed your questions.

---

> > ### Author Response · Authors · 2023-11-20
> > **Thank you for your constructive feedback!**
> >
> > Dear Reviewer, we would like to thank you again for your efforts and time in providing thoughtful feedback and comments. We’ve revised the paper according to your suggestions and replied to all the questions and concerns. Since the discussion period is ending soon, we would greatly appreciate it if you could let us know whether you have any additional comments. Thanks a lot!

---

> ### Comment · Reviewer_FoVN · 2023-11-22
> **Satisfied with answers**
>
> I am satisfied with the authors' responses to my questions and maintain my score. Thank you for the detailed response!

---

### Author Response · Authors · 2023-11-15
**General Response to All Reviewers**

We thank all the reviewers for the very valuable and constructive comments. We have revised the paper as suggested by the reviewers (and colored the revisions in **blue**), and have addressed their comments point-by-point below (please see each individual response). Please do not hesitate to let us know if there are any other questions and comments.

>  **Motivation**: We want to emphasize that the biggest motivation for “fleet learning” is (a) efficiency in data communication and (b) respecting privacy considerations. In Natural Language Processing and Computer Vision, the internet contains large corpuses of easy-to-access data. We do not have similarly large available corpuses yet in robotics. Even if we did, collecting and sharing these raw robot observations and interactions, which may include video and 3D observations, can generate enormous amounts of data that is more challenging to transmit than images or text. We therefore propose that, by sharing *model parameters* rather than by *pooling these massive datasets*, we can improve communication efficiency. Privacy concerns can also arise in many  applications, such as when one robotic manufacturer is  deployed in warehouses or factories for multiple client companies. In this case, the robotics company may not be *allowed* to directly transmit local client data to its centralized model. It can, however, export model weights back to the central training algorithm, and use local data from each client to finetune the centralized model. This is precisely what we are advocating. Ultimately, this merged model can correspond to a large, pretrained so-called “foundation model”; e.g. a future iteration of RT-X (as proposed by Reviewer 2Z2Z).

>  **Experiments**: In addition to the gains in performance shown in various robotic imitation tasks (notably,  Figure 3(d) with alpha = .01 corresponds to a *20% gain* in performance,)  the Appendix provides results supervised learning and linear control experiments. In these cases,  we also find our algorithm to be *significantly outperforming* the baselines. We also conduct new online RL experiments (as suggested by Reviewer DxkE) and find similar improvements. Specifically we use REINFORCE on Openai Gym environment for its training speed and simplicity. We observe that the simplest fleet learning algorithm is already robust to very infrequent communications.

---

### Public Comment · ~Zhiwei_Jia1 · 2023-11-23
**Related Work**

Hi authors. I like your work and think it would be great to add the following work [1], which also utilizes a strategy of population-based training and combining the learned agents for better policy learning, to the related work. Thanks.

[1] Improving Policy Optimization with Generalist-Specialist Learning

---

### Meta-Review · Area_Chair_ZXRN · 2023-12-11

**Metareview:**

The paper introduces 'Fleet-Merge,' a novel method designed to tackle the complexities of policy learning from a network of distributed robots, specifically focusing on issues related to data transmission, such as privacy and communication costs. The method is geared towards a scenario that holds significant practical applications. Despite some concerns raised by the reviewers, the contribution of 'Fleet-Merge' appears to be substantial and robust enough to meet the standards of ICLR. Given the solid nature of the contribution and its relevance to the field, the area chair recommends accepting the submission for publication. This endorsement reflects the belief that 'Fleet-Merge' not only addresses a pertinent challenge in the realm of distributed robotics but also makes a meaningful advancement in policy learning methodologies.

**Justification For Why Not Higher Score:**

As an early work, there are still concerns not reflected in the simple environments tested by this work.

**Justification For Why Not Lower Score:**

The contribution seems to be sufficient to be accepted by ICLR.

---

### Decision · Program_Chairs · 2024-01-16

Accept (poster)